# Clustering with Non-adaptive Subset Queries

**Hadley Black**

UC San Diego

**Euiwoong Lee**

University of Michigan

**Arya Mazumdar**

UC San Diego

**Barna Saha**

UC San Diego

## Abstract

Recovering the underlying clustering of a set $U$ of $n$ points by asking pair-wise *same-cluster* queries has garnered significant interest in the last decade. Given a query $S \subset U, |S| = 2$, the oracle returns *yes* if the points are in the same cluster and *no* otherwise. We study a natural generalization of this problem to *subset queries* for $|S| > 2$, where the oracle returns the number of clusters intersecting $S$. Our aim is to determine the minimum number of queries needed for exactly recovering an arbitrary $k$-clustering. We focus on non-adaptive schemes, where all the queries are asked in one round, thus allowing for the querying process to be parallelized, which is a highly desirable property.

For *adaptive* algorithms with pair-wise queries, the complexity is known to be $\Theta(nk)$, where $k$ is the number of clusters. In contrast, *non-adaptive* pair-wise query algorithms are extremely limited: even for $k = 3$, such algorithms require $\Omega(n^2)$ queries, which matches the trivial $O(n^2)$ upper bound attained by querying every pair of points. Allowing for *subset queries* of unbounded size, $O(n)$ queries is possible with an adaptive scheme. However, the realm of non-adaptive algorithms remains completely unknown. Is it possible to attain algorithms that are *non-adaptive* while still making a near-linear number of queries?

In this paper, we give the first non-adaptive algorithms for clustering with subset queries. We provide, (i) a non-adaptive algorithm making $O(n \log^2 n \log k)$ queries which improves to $O(n \log k)$ when the cluster sizes are within any constant factor of each other, (ii) for constant $k$, a non-adaptive algorithm making $O(n \log \log n)$ queries. In addition to non-adaptivity, we take into account other practical considerations, such as enforcing a bound on query size. For constant $k$, we give an algorithm making $\widetilde{O}(n^2/s^2)$ queries on subsets of size at most $s \leq \sqrt{n}$, which is optimal among all non-adaptive algorithms within a $\log n$-factor. For arbitrary $k$, the dependence varies as $\tilde{O}(n^2/s)$.

## 1 Introduction

Clustering is one of the most fundamental problems in unsupervised machine learning, and permeates beyond the boundaries of statistics and computer science to social sciences, economics and so on. The goal of clustering is to partition items so that similar items are in the same group. The applications of clustering are manifold. However, finding the underlying clusters is sometimes hard for an automated process due to data being noisy, incomplete, but easily discernible by humans. Motivated by this scenario, in order to improve the quality of clustering, early works have studied the so-called clustering under "limited supervision" (e.g.,[1, 2]). Balcan and Blum initiated the study of clustering under active feedback [3] where given the current clustering solution, the users can provide feedback whether a cluster needs to be merged or split. Perhaps a simpler query model would be where users only need to answer the number of clusters, and that too only on a subset of points without requiring to analyze the entire clustering. This scenario is common in unsupervised learning problems, where a centralized algorithm aims to compute a clustering by crowdsourcing. The crowd-workers play the role of an oracle here, and are able to answer simple queries that involve a small subset of the universe.

38th Conference on Neural Information Processing Systems (NeurIPS 2024).

Mazumdar and Saha [4, 5, 6], and in independent works Mitzenmacher and Tsourakis [7], as well as Asthani, Kushagra and Ben-David [8] initiated a theoretical study of clustering with pair-wise aka *same-cluster* queries. Given any pair of points $u, v$, the oracle returns whether $u$ and $v$ belong to the same cluster or not. Such queries are easy to answer and lend itself to simple implementations [9]. This has been subsequently extremely well-studied in the literature, e.g. [10, 11, 4, 12, 13]. In fact, triangle-queries have also been studied, e.g. [14]. Moreover, clustering with pair-wise queries is intimately related to several well-studied problems such as correlation clustering [15, 16, 17, 10, 18], edge-sign prediction problem [19, 7], stochastic block model [20, 21] etc.

Depending on whether there is an interaction between the learner/algorithm and the oracle, the querying algorithms can be classified as adaptive and non-adaptive [5]. In adaptive querying, the learner can decide the next query based on the answers to the previous queries. An algorithm is called *non-adaptive* if all of its queries can be specified in one-round. Non-adaptive algorithms can parallelize the querying process as they decide the entire set of queries apriori. This may greatly speed up the algorithm in practice, significantly reducing the time to acquire answers [22]. Thus, in a crowdsourcing setting being non-adaptive is a highly desirable property. On the flip side, this makes non-adaptive algorithms significantly harder to design. In fact, when adaptivity is allowed, $nk$ pair-wise queries are both necessary and sufficient to recover the entire clustering, where $n$ is the number of points in the ground set to be clustered and $k$ (unknown) is the number of clusters. However as shown in [5] and our Theorem C.1, even for $k = 3$, even randomized *non-adaptive algorithms can do no better than the trivial $O(n^2)$ upper bound attained by querying all pairs.*

We study a generalization of pair-wise queries to subset queries, where given any subset of points, the oracle returns the number of clusters in it. We consider the problem of recovering an unknown $k$-clustering (a partition) on a universe $U$ of $n$ points via black-box access to a *subset query oracle*. More precisely, we assume that there exists a groundtruth partitioning of $U = \bigsqcup_{i=1}^{k} C_i$, and upon querying with a subset $S \subseteq U$, the oracle returns $q(S) = |\{i : C_i \cap S \neq \emptyset\}|$, the number of clusters intersecting $S$. Considering the limitations of pair-wise queries for non-adaptive schemes, we ask the question if it is possible to use subset queries to design significantly better non-adaptive algorithms.

In addition to being a natural model for interactive clustering, this problem also falls into the growing body of work known as *combinatorial search* [23, 24] where the goal is to reconstruct a hidden object by viewing it through the lens of some indirect query model (such as group testing [25, 26, 24, 27, 28]). The problem is also intimately connected to coin weighing where given a hidden vector $x \in \{0, 1\}^n$, the goal is to reconstruct $x$ using queries of the form $q(S) := \sum_{i \in S} x_i$ for $S \subseteq [n]$. It is known that $\Theta(n/\log n)$ is the optimal number of queries [29, 30, 31], which can be obtained by a non-adaptive algorithm. There are improvements for the case when $\|x\|_1 = d$ for $d \ll n$ [32, 33, 34]. Moreover, there has been significant work on graph reconstruction where the task is to reconstruct a hidden graph $G = (V, E)$ from queries of the form $q(S, T) := |\{(u, v) \in E : u \in S, v \in T\}|$ for subsets $S, T \subseteq V$. [35, 36, 37, 38]. There are also algorithms that perform certain tasks more efficiently than learning the whole graph (sometimes using different types of queries) [39, 40, 41, 42, 43, 44, 45, 46], and quantum algorithms that use fewer queries than classical algorithms [47].

It is not too difficult to show that an algorithm making $O(n \log k)$ queries (Appendix H) is possible for $k$-clustering, while $\Omega(n)$ queries is an obvious information theoretic lower bound since each query returns $\log k$ bits of information and the number of possible $k$-clusterings is $k^n = 2^{n \log k}$. In fact, it is possible to have an algorithm with $O(n)$ query complexity (personal communication, Chakrabarty and Liao). However, both of these algorithms are adaptive, ruling them out for the non-adaptive setting. So far, the non-adaptive setting of this problem remained unexplored.

## 1.1 Results

Our main results showcase the significant strength of using subset queries in the non-adaptive setting. We give randomized algorithms that recover the exact clustering with probability $1 - \delta$, for any arbitrary constant $\delta > 0$ using only near-linear number of subset queries.

**Theorem 1.1.** *(Theorem 2.5, simplified) There is a randomized, non-adaptive $k$-clustering algorithm making $O(n \log^2 n \log k)$ subset queries.*

For constant $k$, this dependency can be further improved.

**Theorem 1.2.** *(Theorem 2.2, simplified)* *There is a randomized, non-adaptive $k$-clustering algorithm making $O(n \log \log n)$ subset queries when $k$ is any constant.*

Note that the algorithm of Theorem 1.2 works for any value of $k$, but its dependence on this parameter is inferior to that of Theorem 1.1 (see the formal version Theorem 2.2 for the exact dependence on $k$). Thus, we state the theorem above for constant $k$ to emphasize the much improved dependence on $n$.

Our algorithms also run in polynomial time, and generalizes to work with queries of bounded size.

**Bounding query size:** Another practical consideration is query size, $s$. Depending on the scenarios, and capabilities of the oracle, it may be easier to handle queries on small subsets. An extreme case is *pair-wise queries* ($s = 2$), where $O(nk)$ pair queries are enough with adaptivity but any non-adaptive algorithm has to use $\Omega(n^2)$ queries even for $k = 3$. Since a subset query on $S$ can be simulated by $\binom{|S|}{2}$ pair queries, we immediately get the following theorem.

**Theorem 1.3.** *(Corollary C.2, restated)* *Any non-adaptive $k$-clustering algorithm that is only allowed to query subsets of size at most $s$ must make at least $\Omega(\min(\frac{n^2}{s^2}, n))$ queries.*

Theorems 1.1 and 1.2 above show that this can be bypassed by allowing larger subset queries. However, some of these queries are of size $\Omega(n)$, and this raises the question, *is there a near-linear non-adaptive algorithm which only queries subsets of size at most $O(\sqrt{n})$?* We answer this in the affirmative, implying that our lower bound is tight in terms of $s$.

**Theorem 1.4** (Theorem A.1, informal)**.** *There is a non-adaptive $k$-clustering algorithm making $O(n \log n \log \log n)$ subset queries of size at most $O(\sqrt{n})$ when $k$ is any constant. For all sufficiently small $s = o(\sqrt{n})$, the algorithm makes $O(\frac{n^2}{s^2} \log n)$ subset queries of size at most $s$.*

The result also extends to arbitrary $k$ with slightly worse dependency on $s$ (Theorem 2.5). Our algorithm for bounded queries from Theorem 1.4 has the additional desirable property of being *sample-based* meaning that each of its queries is a set formed by independent, uniform samples. I.e. the algorithm specifies a query size $t \leq s$, and then receives $(S, q(S))$ where $S$ is formed by $t$ i.i.d. uniform samples from $U$. Being sample-based enables the algorithm to leave the task of curating each query up to the individual answering the query. The algorithm needs only to specify the query sizes, and then recover the clustering once the queries have been curated and answered.

**The "roughly balanced" case:** Next, we consider the natural special case of recovering a $k$-clustering when the cluster sizes are within a constant factor of one another. Informally, let us call such a clustering "roughly balanced".

**Theorem 1.5** (Theorems B.1 and E.1, informal)**.** *There are non-adaptive algorithms for recovering a roughly balanced $k$-clustering which make (a) $O(n \log k)$ subset queries when $k \leq O(\frac{n}{\log^3 n})$, and (b) $O(n \log^2 k)$ subset queries for any $k \leq n$.*

**Allowing two rounds of adaptivity**   Finally, we show if we allow an extra round of adaptivity, then that helps to improve the dependency on the logarithmic factors further. Specifically, we prove the following theorems.

**Theorem 1.6** (Theorems F.1 and F.3, informal)**.** *There is a $2$-round deterministic $k$-clustering algorithm making $O(n \log k)$ subset queries. There is a randomized $2$-round algorithm for recovering a roughly-balanced $k$-clustering making $O(n \log \log k)$ subset queries.*

**Organization:**   The remainder of the paper is organized as follows. In Section 2, we give our main results developing non-adaptive algorithms with near-linear query complexity Theorems 1.1 and 1.2. Our results for sample-based, bounded query algorithms are given in Appendix A. Finally, we prove our results for the balanced setting in Appendix B, our lower bounds in Appendix C, and our results for two-round algorithms in Appendix F.

## 2   Algorithms with Nearly Linear Query Complexity

In this section we describe the algorithms behind our main results, Theorems 1.1 and 1.2, and give formal proofs of their correctness. In Section 2.1 we describe an algorithm making $O(n \log \log n)$ subset queries when the number of clusters $k$ is assumed to be a constant. In general, the dependence on the number of clusters is $O(k \log k)$. In Section 2.2, we give an alternative algorithm with $\widetilde{O}(n)$ query complexity for any $k \leq n$.

## 2.1 An $O(n \log \log n)$ Algorithm for Constant $k$

**Warm Up.** When there are only 2 clusters, there is a trivial non-adaptive algorithm making $O(n)$ *pair queries*: Choose an arbitrary $x \in U$ and query $\{x, y\}$ for every $y \in U$. The set of points $y$ where $q(\{x, y\}) = 1$ form one cluster, and the second cluster is the complement. If we allow one more round of adaptivity, then for 3-clustering we could repeat this one more time and again get an $O(n)$ query algorithm. However, for *non-adaptive* 3-clustering it is impossible to do better than the trivial $O(n^2)$ algorithm (see Theorem C.1). Essentially, this is because in order to distinguish the clusterings $(\{x\}, \{y\}, U \setminus \{x, y\})$ and $(\{x, y\}, \emptyset, U \setminus \{x, y\})$ the algorithm must query $\{x, y\}$ and their are $\binom{n}{2}$ ways to hide this pair. Overcoming this barrier using subset queries require significant new ideas.

Our main ideas are best communicated by focusing on the case of 3-clustering. It suffices to correctly reconstruct the two largest clusters, since the third cluster is just the complement of their union. Let $A, B$ denote the largest, and second largest clusters, respectively. Since $|A| \geq n/3$, it is easy to find: sample a random $x \in U$ and query $\{x, y\}$ for every $y \in U$. The cluster containing $x$ is precisely $\{y \in U : q(\{x, y\}) = 1\}$. With probability at least $1/3$, we have $x \in A$ and so repeating this a constant number of times will always recover $A$. On the other hand, $B$ may be arbitrarily small and in this case the procedure clearly fails to recover it. The first observation is that once we know $A$, we can exploit larger subset queries to explore $U \setminus A$ since $q(S \setminus A) = q(S) - \mathbf{1}(S \cap A \neq \emptyset)$. Importantly, the algorithm is non-adaptive and so the choice of $S$ cannot depend on $A$, but we are still able to exploit this trick with the following two strategies. Let $\delta n = |B|$ denote the size of $B$ and note that this implies $|A| \geq (1 - 2\delta)n$ since the third cluster is of size at most $B$.

*Strategy 1:* Suppose a query $S$ contains exactly one point outside of $A$, i.e. $S \setminus A = \{x\}$. Then, for $y \notin A$, $q(S \cup \{y\}) = q(S)$ iff $x, y$ belong to the same cluster. Thus, we can query $S \cup \{y\}$ for every $y \in U$ to learn the cluster containing $x$. If $S$ is a random set of size $t \approx 1/\delta$, then the probability that $|S \setminus A| = 1$ is at least $t \cdot \delta \cdot (1 - 2\delta)^{t-1} = \Omega(1)$. Of course, we do not know $\delta$, but we can try $t = 2^p$ for every $p \leq \log n$ and one of these choices will be within a factor of 2 from $1/\delta$. This gives us an $O(n \log n)$ query algorithm since we make $n$ queries per iteration.

*Strategy 2:* Suppose $S$ intersects $A$ and contains exactly two points outside of $A$, i.e. $S \setminus A = \{x, y\}$. Then, $q(\{x, y\}) = q(S) - 1$ which tells us whether or not $x, y$ belong to the same cluster. If $x, y$ belong to same cluster, add it to a set $E$, and let $G(U \setminus A, E)$ denote a graph on the remaining points with this set of edges. By transitivity, a connected component in this graph corresponds to a subset of one of the remaining two clusters. In particular, if the induced subgraph, $G[B]$, is connected, then we recover $B$. Moreover, if $S$ is a random set of size $t \approx 1/\delta$, then the probability that two points land in $B$ and the rest land in $A$ is at least $\binom{t}{2} \cdot \delta^2 \cdot (1 - 2\delta)^{t-2} = \Omega(1)$. A basic fact from random graph theory says that after $\approx |B| \ln |B| \leq \delta n \ln n$ occurrences of this, $G[B]$ becomes connected with high probability and so querying $\Omega(\delta n \ln n)$ random $S$ of size $\approx 1/\delta$ will suffice. Again, we try $t = 2^p$ for every $p \leq \log n$, resulting in a total of $\approx n \ln n \sum_p 2^{-p} = O(n \log n)$ queries.

Finally, we can combine strategies (1) and (2) as follows to obtain our $O(n \log \log n)$ query algorithm. The main observation is that the query complexity of strategy (2) improves greatly if we assume that $|B|$ is small enough. If we know that $\delta \leq \frac{1}{\log n}$, then we only need to try $t = 2^p \geq \log n$ and so the query complexity becomes $\approx n \ln n \sum_{p \geq \log \log n} 2^{-p} = O(n)$. On the other hand, if we assume that $\delta > \frac{1}{\log n}$, then in strategy (1) we only need to try $p \leq \log \log n$ yielding a total of $O(n \log \log n)$ queries. Combining these yields the final algorithm.

**Remark 2.1** (On approximate clustering). *We point out that these ideas can be used to obtain more efficient algorithms for the easier task of correctly clustering a $(1 - \alpha)$-fraction of points. In this setting we can ignore the case of $\delta < \alpha/2$ (recall the definition of $\delta$ above) as this will only result in an incorrect classification of an $\alpha$-fraction of points. Thus, for example, one can employ "strategy 1" above, but only iterate over $p \leq \log(2/\alpha)$, leading to an $O(n \log \frac{1}{\alpha})$ query algorithm. However, in this paper we focus on the more challenging task of recovering the clustering exactly, and leave the possibility of more efficient approximate algorithms as a possible direction of future work.*

**Algorithm.** A full description of the algorithm is given in pseudocode Alg. 1, which is split into two phases: a "query selection phase", which describes how queries are chosen by the algorithm, and a "reconstruction phase" which describes how the algorithm uses the query responses to determine the clustering. Both phases contain a for-loop iterating over all $p \in \{0, 1, \ldots, \log n\}$ where the goal

of the algorithm during the $p$'th iteration is to learn all remaining clusters of size at least $\frac{n}{2k \cdot 2^p}$. This is accomplished by two different strategies depending on whether $p$ is small or large.

When $p \leq \log \log n$, the algorithm samples $O(k \log k)$ random sets $T$ formed by $2^p$ samples from $U$ and makes a query on $T$ and $T \cup \{x\}$ for every $x \in U$ (see lines 5-9 of Alg. 1). Let $\mathcal{R}_p$ be the union of all clusters reconstructed before phase $p$ (i.e., clusters of size at least $\frac{n}{2k \cdot 2^{p-1}}$). If such a $T$ contains exactly one point $z \in T \setminus \mathcal{R}_p$ belonging to an unrecovered cluster, then we can use these queries to learn the cluster containing $z$ (see lines 24-28 of Alg. 1), since for $x \in U \setminus \mathcal{R}_p$, $q(T) = q(T \cup \{x\})$ if and only if $x, z$ belong to the same cluster. Moreover, we show that this occurs with probability $\Omega(1)$ and repeat this $O(k \log k)$ times to ensure that every cluster $C$ where $|C| \in [\frac{n}{2k \cdot 2^p}, \frac{n}{2k \cdot 2^{p-1}})$ is learned with high probability. The total number of queries made during iterations $p \leq \log \log n$ is $O(n \log \log n \cdot k \log k)$.

When $p > \log \log n$, the algorithm queries $O(nk \cdot \frac{\log n}{2^p})$ random sets $T$ again formed by $2^p$ samples from $U$ (see lines 11-14 of Alg. 1). Note that $\sum_{p > \log \log n} 2^{-p} = O(\frac{1}{\log n})$ and so the total number of queries made during these iterations is $O(nk)$.

We now describe the reconstruction phase (see lines 32-37 of Alg. 1). If $T$ contains exactly two points $x, y \in T \setminus \mathcal{R}_p$ belonging to unrecovered clusters, then we can use the fact that we already know the clustering on $\mathcal{R}_p$ to tell whether or not $x, y$ belong to the same cluster or not, i.e. we can compute $q(\{x, y\}) \in \{1, 2\}$ from $q(T)$. We then consider the set of all such pairs where $q(\{x, y\}) = 1$ (this is $Q''_p$ defined in line 34) and consider the graph $G$ with this edge set, and vertex set $U \setminus \mathcal{R}_p$, the set of points whose cluster hasn't yet been determined. If two points belong to the same connected component in this graph, then they belong to the same cluster. Thus, the analysis for this iteration boils down to showing that with high probability, the induced subgraph $G[C]$ will be connected for every $C$ where $|C| \in [\frac{n}{2k \cdot 2^p}, \frac{n}{2k \cdot 2^{p-1}})$. This is accomplished by applying a basic fact from the theory of random graphs, namely Fact 2.4.

**Analysis**   We restate the main theorem for this section.

**Theorem 2.2.** *There is a non-adaptive algorithm for $k$-clustering that uses $O(n \log \log n \cdot k \log k)$ subset queries and succeeds with probability at least $1 - \delta$ for any constant $\delta > 0$[1].*

The following Lemma 2.3 establishes that after the first $p$ iterations of the algorithm's query selection and reconstruction phases, all clusters of size at least $\frac{n}{2k \cdot 2^p}$ have been learned with high probability. This is the main technical component of the proof. After stating the lemma we show it easily implies that Alg. 1 succeeds with probability at least $99/100$ by an appropriate union bound. The choice of $99/100$ is arbitrary, and can be made $1 - \delta$ for any constant $\delta$.

**Lemma 2.3.** *For each $p = 0, 1, \ldots, \log n$, let $\mathcal{E}_p$ denote the event that all clusters of size at least $\frac{n}{2k \cdot 2^p}$ have been successfully recovered immediately following iteration $p$ of Alg. 1. Then,*

$$\Pr[\neg \mathcal{E}_0] \leq \frac{1}{100k} \quad and \quad \Pr[\neg \mathcal{E}_p \mid \mathcal{E}_{p-1}] \leq \frac{1}{100k} \quad for \ all \ p \in \{1, 2 \ldots, \log n\}.$$

**Proof of Theorem 2.2:**   Before proving Lemma 2.3, we first observe that it immediately implies the correctness of Alg. 1 and thus proves Theorem 2.2. Let $I_0 = (\frac{n}{2k}, n]$ and for $1 \leq p \leq \log n$, let $I_p = [\frac{n}{2k \cdot 2^p}, \frac{n}{2k \cdot 2^{p-1}})$. If there are no clusters $C$ for which $|C| \in I_p$, then trivially $\Pr[\neg \mathcal{E}_p \mid \mathcal{E}_{p-1}] = 0$, and otherwise $\Pr[\neg \mathcal{E}_p \mid \mathcal{E}_{p-1}] \leq \frac{1}{100k}$ by the lemma. Since there are $k$ clusters, clearly there are at most $k$ values of $p$ for which there exists a cluster with size in the interval $I_p$. Using this observation and a union bound, we have

$$\Pr[\neg \mathcal{E}_{\log n}] \leq \Pr[\neg \mathcal{E}_0] + \sum_{p=1}^{\log n} \Pr[\neg \mathcal{E}_p \mid \mathcal{E}_{p-1}] \leq \frac{1}{100}$$

which completes the proof of correctness since the algorithm succeeds iff $\mathcal{E}_{\log n}$ occurs.

**Query complexity:**   During iterations $p < \log \log n$ the algorithm makes at most $O(n \log \log n \cdot k \log k)$ queries. During iterations $p > \log \log n$, it makes at most $O(nk \log n) \sum_{p > \log \log n} 2^{-p} = O(nk)$ queries since $k \leq n$.

---

[1]For simplicity of exposition, we use a constant $\delta$ in our proofs. The success probability can be boosted to any $1 - \frac{1}{\text{poly}(n)}$ by paying a $\log n$ factor in the query complexity in all algorithms.

**Time complexity:** We assume that obtaining a uniform random sample from a set of size $n$ can be done in $O(1)$ time. Thus, since the algorithm makes $O(n \log \log n \cdot k \log k)$ queries and each is on a set of size at most $n$, the total runtime of the query selection phase (lines 3-15) is bounded by $O(n^2 \log \log n \cdot k \log k)$. We now account for the runtime in the reconstruction phase. Lines (25-28) clearly can be performed in $O(n)$ time and so the time spent in lines (24-28) is $O(|Q_p| \cdot n)$. Now, for $T \in Q_p$, checking if $|T \setminus \mathcal{R}_p| = 2$ can clearly be done in $O(n)$ time and so lines (33-34) run in time $O(|Q_p| \cdot n)$. Line (36) amounts to finding every connected component in $G_p$ which can be done in time $O(|Q_p''| + n) = O(|Q_p| + n)$ by iteratively running a BFS (costing time linear in the number of edges plus the number of vertices). Thus, the runtime of the $p$'th iteration of the for-loop is always dominated by $O(|Q_p| \cdot n)$. Since the total number of queries is $O(n \log \log n \cdot k \log k)$, the total runtime of the reconstruction phase is $O(n^2 \log \log n \cdot k \log k)$.

We now prove the main Lemma 2.3.

*Proof. of Lemma 2.3.* Let $\mathcal{C}_p$ denote the set of clusters recovered before phase $p$ and let $\mathcal{R}_p = \bigcup_{C \in \mathcal{C}_p} C$. When $p = 0$, both of these sets are empty. We will consider three cases depending on the value of $p$.

**Case 1:** $p = 0$. Let $C$ denote some cluster of size $|C| \geq \frac{n}{2k}$. Note that in this iteration the sets $T$ sampled by the algorithm in line (7) are singletons. We need to argue that one of these singletons will land in $C$, and thus $C$ is recovered in line (28), with probability at least $1 - \frac{1}{100k^2}$. Since there are at most $k$ clusters, applying a union bound completes the proof in this case.

A uniform random element lands in $C$ with probability at least $\frac{1}{2k}$ and so this fails to occur for all $|Q_0| \geq 4k \ln 10k$ samples with probability at most $(1 - \frac{1}{2k})^{4k \ln 10k} \leq \exp(-2 \ln 10k) = \frac{1}{100k^2}$, as claimed.

**Case 2:** $1 \leq p \leq \log \log n$. Let $C$ denote some cluster with size $|C| \in [\frac{n}{2k \cdot 2^p}, \frac{n}{2k \cdot 2^{p-1}})$. Note that we are conditioning on the event that every cluster of size $\geq \frac{n}{2k \cdot 2^{p-1}}$ has already been successfully recovered after iteration $p - 1$. Thus, the number of elements belonging to unrecovered clusters is $|U \setminus \mathcal{R}_p| \leq k \cdot \frac{n}{2k \cdot 2^{p-1}} = \frac{n}{2^p}$. We need to argue that the set $Q_p$ will contain some $T$ sampled in line (7) such that $T \setminus \mathcal{R}_p = \{z\}$ where $z \in C$, and thus $C$ is successfully recovered in line (28), with probability at least $1 - \frac{1}{100k^2}$. Once this is established, the lemma again follows by a union bound. We have

$$\Pr_{T \,:\, |T|=2^p}[|T \setminus \mathcal{R}_p| = 1 \text{ and } T \setminus \mathcal{R}_p \subseteq C] = |T| \cdot \frac{|C|}{n} \cdot \left(\frac{|\mathcal{R}_p|}{n}\right)^{|T|-1} \geq \frac{2^p}{k \cdot 2^{p+1}} \left(1 - \frac{1}{2^p}\right)^{2^p} \geq \frac{1}{2ek}$$

and so the probability that this occurs for some $T \in Q_p$ is at least $1 - (1 - \frac{1}{2ek})^{4ek \ln 10k} \geq 1 - \frac{1}{100k^2}$, as claimed.

**Case 3:** $p > \log \log n$. Let $C$ denote some cluster with size $|C| \in [\frac{n}{2k \cdot 2^p}, \frac{n}{2k \cdot 2^{p-1}})$. Note that $|U \setminus \mathcal{R}_p| \leq k \cdot \frac{n}{2k \cdot 2^{p-1}} = \frac{n}{2^p}$. Recall from lines (34-35) the definition of $Q_p''$ and recall that $G_p$ is the graph with vertex set $U \setminus \mathcal{R}_p$ and edge set $Q_p''$. We need to argue that the induced subgraph $G_p[C]$ is connected, and thus $C$ is successfully recovered in lines (36-37), with probability at least $1 - \frac{1}{100k^2}$. Once this is established, the lemma again follows by a union bound. We rely on the following standard fact from the theory of random graphs. For completeness, we give a proof in Appendix D.2.

**Fact 2.4.** *Let $G(N, p)$ denote an Erdös-Rényi random graph. That is, the graph contains $N$ vertices and there is an edge between each pair of vertices with probability $p$. If $p \geq 1 - (\delta/3N)^{2/N}$, then $G(N, p)$ is connected with probability at least $1 - \delta$.*

Consider any $x, y \in C$ and observe that

$$\Pr_{T \,:\, |T|=2^p}[T \setminus \mathcal{R}_p = \{x, y\}] = \binom{2^p}{2} \cdot \frac{1}{n^2} \cdot \left(\frac{|\mathcal{R}_p|}{n}\right)^{2^p-2} \geq \frac{2^{2p}}{3n^2} \left(1 - \frac{1}{2^p}\right)^{2^p} \geq \frac{2^{2p}}{10n^2}.$$

---

**Algorithm 1:** Non-adaptive Algorithm for Constant $k$

---

**1 Input:** Subset query access to a hidden partition $C_1 \sqcup \cdots \sqcup C_k = U$ of $|U| = n$ points;

**2** *(Query Selection Phase)*

**3 for** $p = 0, 1, \ldots, \log n$ **do**

**4**      Initialize $Q_p \leftarrow \emptyset$;

**5**      **if** $p \leq \log \log n$ **then**

**6**          **Repeat** $4ek \ln(10k)$ times;

**7**          $\longrightarrow$ Sample $T \subseteq U$ formed by $2^p$ independent uniform samples from $U$;

**8**          $\longrightarrow$ **Query** $T$ and $T \cup \{x\}$ for all $x \in U$;

**9**          $\longrightarrow$ Add $T$ to $Q_p$;

**10**      **end**

**11**      **if** $p > \log \log n$ **then**

**12**          **Repeat** $\frac{40nk \ln(300nk^2)}{2^p}$ times;

**13**          $\longrightarrow$ Sample $T \subseteq U$ formed by $2^p$ independent uniform samples from $U$;

**14**          $\longrightarrow$ **Query** $T$ and add it to $Q_p$;

**15**      **end**

**16 end**

**17** *(Reconstruction Phase)*

**18** Initialize learned cluster set $\mathcal{C}_0 \leftarrow \emptyset$;

**19 for** $p = 0, 1, \ldots, \log n$ **do**

**20**      Let $\mathcal{C}_p$ denote the collection of clusters reconstructed before iteration $p$;

**21**      Let $\mathcal{R}_p = \bigcup_{C \in \mathcal{C}_p} C$ denote the points belonging to these clusters;

**22**      Initialize $\mathcal{C}_{p+1} \leftarrow \mathcal{C}_p$;

**23**      **if** $p \leq \log \log n$ **then**

**24**          **for** $T \in Q_p$ **do**

**25**              **if** $|T \setminus \mathcal{R}_p| = 1$ **then**

**26**                  Let $z$ denote the unique point in $T \setminus \mathcal{R}_p$;

**27**                  If $x \in U \setminus \mathcal{R}_p$, then $q(T) = q(T \cup \{x\})$ iff $x, z$ are in the same cluster;

**28**                  Thus, we add $\{x \in U \setminus \mathcal{R}_p : q(T) = q(T \cup \{x\})\}$ to $\mathcal{C}_{p+1}$;

**29**              **end**

**30**          **end**

**31**      **end**

**32**      **if** $p > \log \log n$ **then**

**33**          Let $Q'_p = \{T \setminus \mathcal{R}_p : T \in Q_p \text{ and } |T \setminus \mathcal{R}_p| = 2\}$. Since each $T \in Q_p$ is a uniform random set, the elements of $Q'_p$ are uniform random pairs in $U \setminus \mathcal{R}_p$;

**34**          Let $Q''_p = \{\{x, y\} \in Q'_p : q(\{x, y\} = 1)\}$ denote the set of pairs in $Q'_p$ where both points lie in the same cluster. This set can be computed since $q(T \setminus \mathcal{R}_p) = q(T) - q(T \cap \mathcal{R}_p)$ and $q(T \cap \mathcal{R}_p)$ is known since at this point we have reconstructed the clustering on $\mathcal{R}_p$;

**35**          Let $G_p$ denote the graph with vertex set $U \setminus \mathcal{R}_p$ and edge set $Q''_p$;

**36**          Let $C_1, \ldots, C_\ell$ denote the connected components of $G_p$ with size at least $\frac{n}{2k \cdot 2^p}$;

**37**          Add $C_1, \ldots, C_\ell$ to $\mathcal{C}_{p+1}$;

**38**      **end**

**39 end**

**40 Output** clustering $\mathcal{C}_{\log n + 1}$

---

Recall that the algorithm queries $|Q_p| = \frac{40 \cdot nk \ln(300nk^2)}{2^p}$ random sets $T$ of size $2^p$. Thus,

$$\Pr_{Q_p}[(x,y) \in E(G_p[C])] = \Pr_{Q_p}\left[\{x,y\} \in Q_p''\right] = \Pr_{Q_p}[\exists T \in Q_p : T \setminus \mathcal{R}_p = \{x,y\}]$$

$$\geq 1 - \left(1 - \frac{2^{2p}}{10n^2}\right)^{40\frac{n}{2^p} \cdot k \ln(300nk^2)}$$

$$\geq 1 - \exp\left(-\frac{2^p}{n} \cdot 4k \ln(300nk^2)\right)$$

and using $|C| \geq \frac{n}{2k \cdot 2^p}$ and $|C| \leq n$, we obtain

$$\Pr_{Q_p}[(x,y) \in E(G_p[C])] \geq 1 - \exp\left(-\frac{2 \ln(300nk^2)}{|C|}\right)$$

$$\geq 1 - \exp\left(-\frac{2 \ln(300k^2|C|)}{|C|}\right) = 1 - \left(\frac{1}{300k^2|C|}\right)^{\frac{2}{|C|}}.$$

Thus, $(x,y)$ is an edge in $G_p[C]$ with probability at least $1 - \left(\frac{1}{300k^2|C|}\right)^{\frac{2}{|C|}}$ and so by Fact 2.4 $G_p[C]$ is connected with probability at least $1 - \frac{1}{100k^2}$, as claimed. $\square$

**Bounded Query Size**   We can restrict the query size to $s \leq \sqrt{n}$, and still achieve a near-linear query complexity. We sketch the main ideas here for the case of $k = 3$ similar to the "warm-up" in Section 2.1. Details are provided in Appendix A. Our Theorem 1.4 gives an $O(n \log n \log \log n)$ query non-adaptive sample-based algorithm using subset queries of size at most $O(\sqrt{n})$. The main idea is to employ "Strategy 2" described in the warm-up section of Section 2.1 with a slight alteration. Let $A, B$ denote the largest, and second largest clusters, respectively, where $|B| = \delta n$ and so $|A| \geq (1 - 2\delta)n$. Observe that if we take a random set $S$ of size $t \approx \sqrt{1/\delta}$, then the probability that two points land in $B$ and the rest land in $A$ is at least $\binom{t}{2} \cdot \delta^2 \cdot (1 - 2\delta)^{t-2} = \Omega(\delta)$. Recalling the definition of the graph $G$ and the discussion in Section 2.1, after querying $\Omega(n \ln n)$ such $S$, the induced subgraph $G[B]$ becomes connected with high probability, thus recovering the clustering. Similar ideas let us generaize to any $s$, and achieve an optimal dependency on $s$ as stated in Corollary C.2 for constant $k$.

## 2.2   An $O(n \log^2 n \log k)$ Algorithm for General $k$

We now consider the situation with general $k$, for which our algorithm and analysis follow a completely different approach by using techniques from combinatorial group testing.

**Warm up.**   The main subroutine in our algorithm is a procedure for recovering the support of a Boolean vector via OR queries. Given a vector $v \in \{0,1\}^n$, an OR query on a set $S \subseteq [n]$ returns $\mathsf{OR}_S(v) = \bigvee_{i \in S} v_i$, i.e. it returns 1 iff $v$ has a 1-valued coordinate in $S$. The problem of recovering the support of $v$, $\mathsf{supp}(v) = \{i \colon v_i = 1\}$ via OR queries is a basic problem from the group testing and coin-weighing literature. The relevance of this problem for $k$-clustering with subset queries is as follows. Consider a hidden clustering $C_1 \sqcup \cdots \sqcup C_k = U$. Given $x \in U$, let $C(x)$ denote the cluster containing $U = \{x_1, \ldots, x_n\}$ (an arbitrary ordering of $U$), and let $v^{(x)} \in \{0,1\}^n$ denote the Boolean vector where $v_i^{(x)} = \mathbf{1}(x_i \in C(x))$. An OR query on set $S$ to $v^{(x)}$ can be simulated by a subset query to the clustering on sets $S$ and $S \cup \{x\}$ since

$$\mathsf{OR}_S(v^{(x)}) = \bigvee_{i \in S} v_i^{(x)} = \mathbf{1}(C(x) \cap S \neq \emptyset) = \mathbf{1}(q(S \cup \{x\}) = q(S)).$$

Thus, the problem or reconstructing $C(x)$ via subset queries is equivalent to the problem of recovering $v^{(x)}$ via OR queries, up to a factor of 2 in the query complexity.

Then, to learn a cluster $C$ with size $\frac{n}{2^p} \leq |C| \leq \frac{n}{2^{p-1}}$ it suffices to sample $O(2^p)$ random $x$ (one of which lands in $C$ with high probability) and then recover $C(x)$ using $O(\frac{n}{2^p} \log \frac{n}{\delta})$ OR queries. Iterating over every $p \leq \log n$ and boosting the number of samples to guarantee a high probability of success for all $k$ clusters yields our algorithm.

This algorithm can also be restricted to only make subset queries of size at most $s$, and the query complexity scales with $\frac{1}{s}$.

**Theorem 2.5.** *For every* $s \in [2, n]$*, there is a non-adaptive* $k$*-clustering algorithm making* $O(n \log n \log k \cdot (\frac{n}{s} + \log s))$ *subset queries of size at most* $s$*. In particular, for unbounded query size the algorithm makes* $O(n \log^2 n \log k)$ *queries.*

**Proof of Theorem 2.5**  We will use the following lemma for recovering $\mathsf{supp}(v) = \{i \colon v_i = 1\}$ via OR queries. We prove and discuss this lemma in Appendix D.1 (see Lemma D.5).

**Lemma 2.6.** *Let* $v \in \{0, 1\}^n$ *and* $s, t \geq 1$ *be positive integers where* $s \leq \frac{n}{t}$*. There is a non-adaptive algorithm that makes* $O(\frac{n}{s} \log \frac{n}{\delta})$ OR *queries on subsets of size* $s$*, and if* $|\mathsf{supp}(v)| \leq t$*, returns* $\mathsf{supp}(v)$ *with probability* $1 - \delta$*, and otherwise certifies that* $|\mathsf{supp}(v)| > t$*. The algorithm runs in time* $O(n \log \frac{n}{\delta})$*.*

Recall that $\mathsf{OR}_S(v^{(x)}) = \mathbf{1}(q(S \cup \{x\}) = q(S))$, i.e. an OR query on $S$ is simulated by subset queries on sets $S$ and $S \cup \{x\}$. Thus, we immediately get the following corollary.

**Corollary 2.7.** *Let* $x \in U$ *and* $r \geq 2, t \geq 1$ *be positive integers where* $r \leq \frac{n}{t}$*. There is a non-adaptive algorithm that makes* $O(\frac{n}{r} \log \frac{n}{\delta})$ *subset-queries on sets of size at most* $r$*, and if* $|C(x)| \leq t$*, returns* $C(x)$ *with probability* $1 - \delta$*, and otherwise certifies that* $|C(x)| > t$*. The algorithm runs in time* $O(n \log \frac{n}{\delta})$*.*

**Algorithm**  The pseudocode for the algorithm is given in Alg. 2. The idea is to draw random points $x \in U$ (line 5) and then use the procedure from Corollary 2.7 as a subroutine to try to learn $C(x)$ (line 6). By the corollary, this will succeed with high probability in recovering $C(x)$ as long as $t$ is set to something larger than $|C(x)|$. Note that the query complexity of this subroutine depends[2] on $t$. If a cluster $C$ is small, then $\Pr[x \in C]$ is small, but we can call the subroutine with small $t$, while if $C(x)$ is large, then $\Pr[x \in C]$ is reasonably large, though we will need to call the subroutine with larger $t$. Concretely, the algorithm iterates over every $p \in \{1, \ldots, \log n\}$ (line 3), and in iteration $p$ the goal is to learn every cluster $C$ with $|C| \in [\frac{n}{2^p}, \frac{n}{2^{p-1}}]$. To accomplish this, we sample $\Theta(2^p \log k)$ random points $x \in U$ (line 4-5) and for each one, call the subroutine with $t = \frac{n}{2^{p-1}}$ (line 6), which is an upper bound on the sizes of the clusters we are trying to learn. Note that we always invoke the corollary with query size $r = \min(s, 2^{p-1}) \leq s$, enforcing the query size bounded stated in Theorem 2.5.

---

**Algorithm 2:** Non-adaptive Algorithm for General $k$

---
1 **Input:** Subset query access to a hidden partition $C_1 \sqcup \cdots \sqcup C_k = U$ of $|U| = n$ points;
2 Initialize hypothesis clustering $\mathcal{C} \leftarrow \emptyset$;
3 **for** $p = 1, \ldots, \log n$ **do**
4 $\quad$ **Repeat** $2^p \ln(200k)$ times:
5 $\quad$ $\longrightarrow$ Sample $x \in U$ uniformly at random;
6 $\quad$ $\longrightarrow$ Run the procedure from Corollary 2.7 on $x$ with $t = \frac{n}{2^{p-1}}$, query-size $r = \min(s, 2^{p-1})$,
$\quad\quad$ and error probability $\delta = \frac{1}{200k}$. This outputs $C(x)$, the cluster containing $x$, with
$\quad\quad$ probability at least $1 - \delta$ if $|C(x)| \leq t$;
7 $\quad$ $\longrightarrow$ If the procedure returns a set $C$, then set $\mathcal{C} \leftarrow \mathcal{C} \cup \{C\}$. Otherwise, continue;
8 **end**
9 **Output** the clustering $\mathcal{C}$.

---

**Query complexity:**  Note that the number of queries made in line (6) during the $p$'th iteration is $O(\frac{n}{s} \log n)$ when $2^{p-1} \geq s$, and $O(\frac{n}{2^p} \log n)$ when $2^{p-1} < s$. Therefore, the total number of queries made is at most

$$O(\log k) \left( \sum_{p \colon 1 \leq 2^{p-1} < s} O\left(2^p \cdot \frac{n}{2^p} \log n\right) + \sum_{p \colon s \leq 2^{p-1} \leq n} O\left(2^p \cdot \frac{n}{s} \log n\right) \right).$$

The first sum is bounded by $O(n \log n \log s)$ and the second sum is bounded by $O(\frac{n^2}{s} \log n)$. The time-complexity is clearly identical by Corollary 2.7.

---

[2]For intuition, if the subroutine is called with $r = \frac{n}{t}$, then Corollary 2.7 makes $O(t \log \frac{n}{\delta})$ queries.

**Time complexity:** We assume that attaining a uniform sample from a set of size $n$ can be performed in $O(1)$ time. The procedure in line (6) has runtime at most $O(n \log n)$ since we set $\delta = \Theta(\frac{1}{k})$. Thus, the total runtime of the algorithm is $O(n \log n \log k) \cdot \sum_{p \leq \log n} 2^p = O(n^2 \log n \log k)$.

**Correctness:** Consider any cluster $C$ and let $p \in \{1, \ldots, \log n\}$ be such that $\frac{n}{2^p} \leq |C| \leq \frac{n}{2^{p-1}}$. Let $\mathcal{E}_C$ denote the event that some element $x \in C$ is sampled in line (5) during iteration $p$. Let $\mathcal{R}_C$ denote the event that $C \in \mathcal{C}$ when the algorithm terminates. Observe that by Corollary 2.7, $\Pr[\mathcal{R}_C \mid \mathcal{E}_C] \geq 1 - \delta = 1 - \frac{1}{200k}$. Moreover, using our lower bound on $C$ we have

$$\Pr[\neg \mathcal{E}_C] \leq \left(1 - \frac{|C|}{n}\right)^{2^p \ln 200k} \leq \left(1 - \frac{1}{2^p}\right)^{2^p \ln 200k} \leq \frac{1}{200k}.$$

Thus, $\Pr[\neg \mathcal{R}_C] \leq \Pr[\neg \mathcal{E}_C] + \Pr[\neg \mathcal{R}_C \mid \mathcal{E}_C] \leq \frac{1}{100k}$ and taking another union bound over all $k$ clusters completes the proof.

**Acknowledgements.** Hadley Black, Arya Mazumdar, and Barna Saha were supported by NSF TRIPODS Institute grant 2217058 (EnCORE) and NSF 2133484. Euiwoong Lee was also supported in part by NSF grant 2236669 and Google. The collaboration is the result of an EnCORE Institute Workshop.

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

# A    Bounded Query Size and Sample-Based Algorithms

In this section we present an algorithm using subset queries with size bounded by $s$, which matches the lower bound of Theorem C.1, up to a $\log n$-factor. Our algorithm has the additional desirable property of being *sample-based*, meaning that the subsets it queries are formed by taking uniform independent samples. In addition to Theorem A.1, we also obtain a non-adaptive sample-based algorithm using $O(nk \log n)$ unbounded queries in Theorem G.1, using a similar approach. We also show a lower bound of $\Omega(n \log n)$ for any $k \geq 2$ in Appendix C.2 for sample-based algorithms, showing that the dependence on $n$ is optimal for this special class of algorithms.

**Theorem A.1.** *There are non-adaptive, sample-based $k$-clustering algorithms making (a) $O(nk \log n \log \log n)$ subset queries of size at most $O(\sqrt{n})$, and (b) $O(\frac{n^2}{s^2} k \log n)$ subset queries of size at most $s = n^{1/2-\delta}$ for any constant $\delta \in (0, 1/2)$. Each algorithm is correct with probability at least $99/100$.*

For convenience, we will parameterize the query-size bound by $s = n^{1/r}$ where $r$ is any positive real number in the range $2 \leq r \leq \log n$. Before proving the theorem formally, we informally describe the algorithm and its analysis. A full description of the algorithm is given in pseudocode in Alg. 3, which is split into two phases: a "query selection phase", describing how queries are chosen by the algorithm, and a "reconstruction phase", describing how the algorithm uses the query responses to determine the clustering. Both phases contain a for-loop iterating over all $p \in \{0, 1, \ldots, \log_r \log n - 1\}$ where the goal of the algorithm during the $p$'th iteration is to learn all remaining clusters of size at least $\frac{n}{k} \cdot 2^{-r^{p+1}}$. We prove that this occurs with high probability in Lemma 2.3, which gives the main analysis. If each iteration is successful in doing so than the entire clustering has been learned successfully after iteration $p = \log_r \log n - 1$ (since $2^{-r^{\log_r \log n}} = 2^{-\log n} = \frac{1}{n}$), and we justify this formally just after the statement of Lemma A.2.

We describe the algorithm and it's analysis informally for the case of $r = 2$, i.e. when the query sizes are bounded by $s = \sqrt{n}$. We also refer the reader to Section 2 for discussion of the ideas for the simple case of $k = 3$. Consider some iteration $p \in \{0, 1, \ldots, \log \log n - 1\}$ and suppose that prior to this iteration, all clusters of size at least $\frac{n}{k} \cdot 2^{-2^p}$ have been successfully recovered. Let $\mathcal{C}_p$ denote the collection of all such clusters and let $\mathcal{R}_p = \bigsqcup_{C \in \mathcal{C}_p} C$ be the set of points they contain. The goal in iteration $p$ is to learn every cluster $C$ with $|C| \in [\frac{n}{k} \cdot 2^{-2^{p+1}}, \frac{n}{k} \cdot 2^{-2^p})$. The algorithm queries $O(nk \log n)$ random sets $T$ formed by $2^{2^p}$ samples[3] from $U$ (see lines 5-7 of Alg. 3). Similar to the proof of Theorem 2.2, if $T$ contains exactly two points $x, y \in T \setminus \mathcal{R}_p$ belonging to unrecovered clusters, then we can use the fact that we already know the clustering on $\mathcal{R}_p$ to tell whether or not $x, y$ belong to the same cluster or not, i.e. we can compute $q(\{x, y\}) \in \{1, 2\}$ from $q(T)$. We then consider the set of all such pairs where $q(\{x, y\}) = 1$ (this is $Q_p''$ defined in line 16) and consider the graph $G$ with this edge set, and vertex set $U \setminus \mathcal{R}_p$, the set of points whose cluster hasn't yet been determined. If two points belong to the same connected component in this graph, then they belong to the same cluster. Thus, the analysis boils down to showing that with high probability, the induced subgraph $G[C]$ will be connected for every $C$ where $|C| \in [\frac{n}{k} \cdot 2^{-2^{p+1}}, \frac{n}{k} \cdot 2^p)$. This is accomplished by applying a basic fact from the theory of random graphs, namely Fact 2.4.

**Proof of Theorem A.1:**    The following Lemma A.2 establishes that after the first $p$ iterations of the algorithm's query selection and reconstruction phases, all clusters of size at least $\frac{n}{k} \cdot 2^{-r^{p+1}}$ have been learned with high probability. This is the main effort of the proof. After stating the lemma we show it easily implies that Alg. 3 succeeds with probability at least $99/100$ by an appropriate union bound.

**Lemma A.2.** *For each $p = 0, 1, \ldots, \log_r \log n - 1$, let $\mathcal{E}_p$ denote the event that all clusters of size at least $\frac{n}{k} \cdot 2^{-r^{p+1}}$ have been successfully recovered immediately following iteration $p$ of Alg. 3. Then,*

$$\Pr[\neg \mathcal{E}_0] \leq \frac{1}{100k} \quad and \quad \Pr[\neg \mathcal{E}_p \mid \mathcal{E}_{p-1}] \leq \frac{1}{100k} \quad for \ all \ p \in \{1, 2, \ldots, \log_r \log n - 1\}.$$

Before proving Lemma A.2, we observe that it immediately implies Theorem A.1 as follows. Let $I_0 = [\frac{n}{k} \cdot 2^{-r}, n]$ and for $1 \leq p < \log_r \log n$, let $I_p = [\frac{n}{k} \cdot 2^{-r^{p+1}}, \frac{n}{k} \cdot 2^{-r^p})$. If there are no clusters

---

[3]Note that $p \leq \log \log n - 1$ and so $2^{2^p} \leq 2^{\frac{1}{2} \log n} = \sqrt{n}$.

**Algorithm 3:** Sample-Based Algorithm Using Bounded Queries

---

1   **Input:** Subset query access to a hidden partition $C_1 \sqcup \cdots \sqcup C_k = U$ of $|U| = n$ points;

2   *(Query Selection Phase)*

3   **for** $p = 0, 1, \ldots, \log_r \log n - 1$ **do**

4      Initialize query set $Q_p \leftarrow \emptyset$;

5      **Repeat** $20 \cdot nk \ln(300nk^2) \cdot 2^{r^{p+1}(1-\frac{2}{r})}$ times;

6      $\longrightarrow$ Sample $T \subseteq U$ formed by $2^{r^p}$ independent uniform samples from $U$;

7      $\longrightarrow$ **Query** $T$ and add it to $Q_p$;

8   **end**

9   *(Reconstruction Phase)*

10   Initialize learned cluster set $\mathcal{C}_0 \leftarrow \emptyset$;

11   **for** $p = 0, 1, \ldots, \log_r \log n - 1$ **do**

12      Let $\mathcal{C}_p$ denote the collection of clusters reconstructed before iteration $p$;

13      Let $\mathcal{R}_p = \bigcup_{C \in \mathcal{C}_p} C$ denote the points belonging to these clusters;

14      Initialize $\mathcal{C}_{p+1} \leftarrow \mathcal{C}_p$;

15      Let $Q'_p = \{T \setminus \mathcal{R}_p : T \in Q_p \text{ and } |T \setminus \mathcal{R}_p| = 2\}$. Since each $T \in Q_p$ is a uniform random set, the elements of $Q'_p$ are uniform random pairs in $U \setminus \mathcal{R}_p$;

16      Let $Q''_p = \{\{x, y\} \in Q'_p : q(\{x, y\} = 1)\}$ denote the set of pairs in $Q'_p$ where both points lie in the same cluster. This set can be computed since $q(T \setminus \mathcal{R}_p) = q(T) - q(T \cap \mathcal{R}_p)$ and $q(T \cap \mathcal{R}_p)$ is known since at this point we have reconstructed the clustering on $\mathcal{R}_p$;

17      Let $G_p$ denote the graph with vertex set $U \setminus \mathcal{R}_p$ and edge set $Q''_p$;

18      Let $C_1, \ldots, C_\ell$ denote the connected components of $G_p$ with size at least $\frac{n}{k} \cdot 2^{-r^{p+1}}$;

19      Add $C_1, \ldots, C_\ell$ to $\mathcal{C}_{p+1}$;

20   **end**

21   **Output** clustering $\mathcal{C}_{\log_r \log n}$;

---

$C$ for which $|C| \in I_p$, then trivially $\Pr[\neg \mathcal{E}_p \mid \mathcal{E}_{p-1}] = 0$, and otherwise $\Pr[\neg \mathcal{E}_p \mid \mathcal{E}_{p-1}] \leq \frac{1}{100k}$ by the lemma. Since there are $k$ clusters, clearly there are at most $k$ values of $p$ for which there exists a cluster with size in the interval $I_p$. Using this observation and a union bound, we have

$$\Pr[\neg \mathcal{E}_{\log_r \log n - 1}] \leq \Pr[\neg \mathcal{E}_0] + \sum_{p=1}^{\log_r \log n} \Pr[\neg \mathcal{E}_p \mid \mathcal{E}_{p-1}] \leq \frac{1}{100}$$

which completes the proof of correctness since the algorithm succeeds iff $\mathcal{E}_{\log_r \log n - 1}$ occurs.

**Query complexity:** Note that the total number of queries made is $O(nk \log n) \cdot \sum_{p=1}^{\log_r \log n} 2^{r^p(1-\frac{2}{r})}$. When $r = 2$, the summation evaluates to $\log \log n$ which establishes the query complexity in item (a) of [Theorem A.1](#).

Otherwise, let $r = 2 + C$ for some constant $C > 0$. We will argue that $2^{r^p(1-\frac{2}{r})} \leq \frac{1}{2} 2^{r^{p+1}(1-\frac{2}{r})}$ for any $p \leq \log_r \log n - 1$ greater than some constant and thus the summation is bounded as

$$\sum_{p=1}^{\log_r \log n} 2^{r^p(1-\frac{2}{r})} = O(2^{r^{\log_r \log n}(1-\frac{2}{r})}) = O(n^{1-\frac{2}{r}}) = O(n/s^2)$$

establishing the query complexity in item (b) of [Theorem A.1](#). Observe that $2^{r^p(1-\frac{2}{r})} \leq \frac{1}{2} 2^{r^{p+1}(1-\frac{2}{r})}$ is equivalent to $r^p(1 - \frac{2}{r}) \leq r^{p+1}(1 - \frac{2}{r}) - 1$, or equivalently

$$r^{p-1} \geq \frac{1}{(r-1)(r-2)} = \frac{1}{C(1+C)}$$

which clearly holds as long as $p - 1 > \log \frac{1}{C}$ since $r > 2$.

**Time complexity:** We assume that sampling a uniform random element from a set of size $n$ can be done in $O(1)$ time. Thus any set that is sampled during the course of the algorithm can be constructed

in $O(s)$ time. No matter the value of $s$, the number of queries made by the algorithm is dominated by $O(\frac{n^2}{s^2} k \log n \log \log n)$. Thus, the runtime of the query selection phase (lines 3-7) is bounded by $O(\frac{n^2}{s} k \log n \log \log n)$. Now for the reconstruction phase. In line (15), $|T \setminus \mathcal{R}_p|$ can be computed in $O(n)$ time and so lines (15-16) take time $O(|Q_p| \cdot n)$. Line (18) amounts to finding every connected component in $G_p$ which can be done in time $O(|Q_p''| + n) = O(|Q_p| + n)$ by iteratively running a BFS (costing time linear in the number of edges plus the number of vertices). Thus, the runtime of the $p$'th iteration of the for-loop is always dominated by $O(|Q_p| \cdot n)$. Since the total number of queries is dominated by $O(\frac{n^2}{s^2} k \log n \log \log n)$, the total runtime of the reconstruction phase (lines 11-19) is $O(\frac{n^3}{s^2} k \log n \log \log n)$, which dominates the runtime of the query selection phase.

We now prove the main Lemma A.2.

*Proof. of Lemma A.2.* Let $\mathcal{R}_p$ denote the set of points belonging to a cluster which has been recovered before iteration $p$.

**Case 1:** $p = 0$. In this iteration, the algorithm queries $|Q_0| \geq 8 \cdot nk \ln(300nk^2) \cdot 2^{r-2}$ random pairs and we need to show that it successfully recovers all clusters with size at least $\frac{n}{k \cdot 2^r}$ with probability at least $1 - \frac{1}{100k}$. Let $C$ denote any such cluster and recall from lines (16-17) the definition of the graph $G_0$ with vertex set $U$ and edge set $Q_0''$. We will show that the induced subgraph $G_0[C]$ is connected, and thus $C$ is correctly recovered in lines (18-19), with probability at least $1 - \frac{1}{100k^2}$. Since there are at most $k$ clusters, the lemma holds by a union bound.

Consider any two vertices $x, y \in C$ and note that $|Q_0| \geq \frac{2n^2 \ln(300nk^2)}{|C|}$ since $|C| \geq \frac{n}{k \cdot 2^r}$. We lower bound the probability that $(x, y)$ is an edge in $G_0[C]$ as follows. Note that this occurs iff $\{x, y\} \in Q_0$. Thus,

$$
\begin{aligned}
\Pr_{Q_0}[(x,y) \in E(G_0[C])] = \Pr_{Q_0}[\{x,y\} \in Q_0] &= 1 - \left(1 - \frac{1}{n^2}\right)^{|Q_0|} \\
&\geq 1 - \exp\left(-\frac{2\ln(300nk^2)}{|C|}\right) \\
&\geq 1 - \exp\left(-\frac{2\ln(300k^2|C|)}{|C|}\right) = 1 - \left(\frac{1}{300k^2|C|}\right)^{\frac{2}{|C|}} \quad (1)
\end{aligned}
$$

and so by Fact 2.4, $G_0[C]$ is connected with probability at least $1 - \frac{1}{100k^2}$ as claimed.

**Case 2:** $1 \leq p < \log_r \log n$. Recall from lines (12-13) that $\mathcal{C}_p$ denotes the set of clusters recovered prior to iteration $p$ and $\mathcal{R}_p = \bigcup_{C \in \mathcal{C}_p} C$ is the set of points belonging to these clusters. Note that we are conditioning on the event that every cluster of size at least $\frac{n}{k} \cdot 2^{-r^p}$ has been recovered prior to iteration $p$. Let $C$ denote some cluster with size

$$
|C| \in \left[\frac{n}{k} \cdot 2^{-r^{p+1}}, \frac{n}{k} \cdot 2^{-r^p}\right) \text{ and note that } |U \setminus \mathcal{R}_p| \leq k \cdot \frac{n}{k} \cdot 2^{-r^p} = n \cdot 2^{-r^p}.
$$

Recall from lines (16-17) the definition of $Q_p''$ and that $G_p$ is the graph with vertex set $U \setminus \mathcal{R}_p$ and edge set $Q_p''$. We need to argue that the induced subgraph $G_p[C]$ is connected, and thus $C$ is correctly recovered in lines (18-19), with probability at least $1 - \frac{1}{100k^2}$. Since there are at most $k$ clusters, a union bound completes the proof of the lemma.

Consider any two vertices $x, y \in C$. We lower bound the probability that $(x, y)$ is an edge in $G_p[C]$, which occurs iff there is some $T \in Q_p$ where $T \setminus \mathcal{R}_p = \{x, y\}$. We have

$$
\Pr_{T:\ |T|=2^{r^p}}[T \setminus \mathcal{R}_p = \{x,y\}] = \binom{2^{r^p}}{2} \cdot \frac{1}{n^2} \cdot \left(\frac{|\mathcal{R}_p|}{n}\right)^{t-2} \geq \frac{2^{2r^p}}{3n^2}\left(1 - 2^{-r^p}\right)^t \geq \frac{2^{2r^p}}{10n^2}
$$

and since $|Q_p| = 20nk\ln(300nk^2) \cdot 2^{r^{p+1}(1-\frac{2}{r})}$, we have

$$\Pr_{Q_p}\left[(x,y) \in E(G_p[C])\right] = \Pr_{Q_p}\left[\{x,y\} \in Q_p''\right] = \Pr_{Q_p}\left[\exists T \in Q_p : T \setminus \mathcal{R}_p = \{x,y\}\right]$$

$$\geq 1 - \left(1 - \frac{2^{2r^p}}{10n^2}\right)^{20 \cdot nk \cdot 2^{r^{p+1}-2r^p}\ln(300nk^2)}$$

$$\geq 1 - \exp\left(-\frac{2 \cdot 2^{r^{p+1}}k\ln(300nk^2)}{n}\right)$$

and plugging in $|C| \geq \frac{n}{k} \cdot 2^{-r^{p+1}}$ and $|C| \leq n$ into the RHS yields

$$\Pr_{Q_p}\left[(x,y) \in E(G_p[C])\right] \geq 1 - \exp\left(-\frac{2\ln(300nk^2)}{|C|}\right)$$

$$\geq 1 - \exp\left(-\frac{2\ln(300k^2|C|)}{|C|}\right) = 1 - \left(\frac{1}{300k^2|C|}\right)^{\frac{2}{|C|}}.$$

Therefore, $(x,y)$ is an edge in $G_p[C]$ with probability at least $1 - \left(\frac{1}{300k^2|C|}\right)^{\frac{2}{|C|}}$, which by Fact 2.4 implies that $G_p[C]$ is connected with probability at least $1 - \frac{1}{100k^2}$ as claimed. $\qquad\square$

## B  The Special Case of Balanced Clusters

Given $B \geq 1$, we say that a $k$-partition $C_1, \ldots, C_k$ is $B$-balanced if $\frac{n}{Bk} \leq |C_i| \leq \frac{Bn}{k}$ for all $i \in [k]$. In this section we prove the following theorem, which gives a non-adaptive algorithm for recovering a roughly balanced $k$-clustering making $O(n \log k)$ subset queries when $k = O(\frac{n}{\log^3 n})$. We give an alternative algorithm making $O(n \log^2 k)$ queries for arbitrary $k$ in Appendix E. We also described a two-round algorithm for this setting making $O(n \log \log k)$ queries in Appendix F.2.

**Theorem B.1.** *There is a non-adaptive algorithm that recovers a $B$-balanced $k$-clustering using $O(B^2 n \log k) + O(Bk \log^4 k)$ subset queries of size $O(k \log k)$ and succeeds with probability $49/50$.*

Pseudocode for the algorithm is given in Alg. 4. In line (3) we draw $s = \Theta(B^2 \log k)$ sets $T_1, \ldots, T_s$ each formed by $k/B$ samples from $U$ and in line (5) learn the clustering over their union using Theorem 2.5. I.e., for $T = T_1 \cup \cdots \cup T_s$, we find $R_j = T \cap C_j$. Then, we query $T_i$ and $T_i \cup \{x\}$ for every $x \in U$ and every $i \in [s]$ in line (5). Now, consider some point $x \in U$ and let $j^*$ be it's cluster's index. Note that $q(T_i \cup \{x\}) = q(T_i)$ iff $T_i$ intersects $C_{j^*}$. Thus, if $T_i$ does not intersect $C_{j^*}$, then every cluster $j$ that $T_i$ intersects can be ruled out as a candidate for being the cluster containing $x$. The set $J_x$ computed in line (8) is the set of all $j$ which can be ruled out in this way. If for every $j \neq j^*$, there is some $T_i$ containing $j$, but not $j^*$, then $J_x = \{j^*\}$ and we determine $j^*$ in line (9). This occurs *for every* $x \in U$ if the following holds: for every pair $(j,j') \in \binom{U}{2}$, there exists $T_i$ intersecting $C_j$, but not $C_{j'}$. We show in Claim B.2 that this happens with high probability.

**Proof of Theorem B.1**  There are $O(B^2 n \log k)$ queries made in line (5) and $O(B \cdot k \log^4 k)$ queries in line (4), since $|\bigcup_{i \in [s]} T_i| = O(Bk \log k)$.

**Time complexity:**  We assume the attaining a uniform sample from any set can be done $O(1)$ time. Thus, constructing sets $T_1, \ldots, T_s$ in line (3) costs $O(Bk \ln k)$ time and by Theorem 2.5 line (4) costs $O(k^2 B^2 \ln^4 k)$. Line (5) costs $O(n \cdot s) = O(B^2 n \ln k)$ time. Constructing $J_x$ in line (8) amounts to checking if $q(T_i \cup \{x\}) \neq q(T_i)$ and if $T_i \cap R_j \neq \emptyset$ for each $i \in [s]$ and $j \in [k]$. This can be done in time $O(|T_i| \cdot |R_j|) = O(k^2 \ln k)$ simply using $|R_j| \leq |R| = O(Bk \ln k)$ and $|T_i| = k/B$. Thus, the total runtime of lines (7-14) is dominated by $O(nk^2 \ln k)$.

**Correctness:**  We now prove correctness, which is due to the following claim.

**Claim B.2.** *For $i \in [s], j \in [k]$, let $\mathcal{E}_{i,j}$ denote the event that $T_i \cap C_j \neq \emptyset$. Then,*

$$\Pr_{T_1, \ldots, T_s}\left[\forall(j,j') \in \binom{[k]}{2}, \exists i \in [s] : \mathcal{E}_{i,j} \wedge \neg\mathcal{E}_{i,j'}\right] \geq \frac{99}{100}. \tag{2}$$

**Algorithm 4:** Algorithm for the $B$-Balanced Case

---

1 **Input:** Subset query access to a $B$-balanced partition $C_1 \sqcup \cdots \sqcup C_k = U$ of $|U| = n$ points;

2 *(Query Selection Phase)*

3 Choose $s = 2eB^2 \ln(100k^2)$ sets $T_1, \ldots, T_s$ each formed by $\frac{k}{B}$ uniform samples from $U$;

4 Run the algorithm from Theorem 2.5 to learn the clustering restricted on $R = \bigcup_{i=1}^{s} T_i$. Let $R_1, \ldots, R_k$ be the output of the algorithm. I.e., if the algorithm is successful, then $R_j = R \cap C_j$ for all $j \in [k]$;

5 **Query** $T_i$ and $T_i \cup \{x\}$ for all $i \in [s]$ and all $x \in U$;

6 *(Reconstruction Phase)*

7 **for** $x \in U$ **do**

8      Let $J_x = \bigcup_{i \in [s]\colon q(T_i \cup \{x\}) \neq q(T_i)} \{j \in [k]\colon T_i \cap R_j \neq \emptyset\}$. Note that $T_i \cap R_j \neq \emptyset$ iff $T_i \cap C_j \neq \emptyset$. Note that $q(T_i \cup \{x\}) \neq q(T_i)$ iff $x$ does not belong to any cluster that is hit by $T_i$. Thus, $J_x$ is the collection of all $j$ such that some set $T_i$ has revealed that $x \notin C_j$;

9      **if** $|J_x| = k - 1$ **then**

10        Add $x$ to $R_{j^*}$ where $j^*$ is the unique element of $[k] \setminus J_x$;

11     **else**

12        **Output** fail;

13     **end**

14 **end**

15 **Output** clustering $(R_1, \ldots, R_k)$;

---

*Proof.* Firstly, for fixed $i \in [s]$ and $j \neq j'$, since each cluster's size is bounded in the interval $[\frac{n}{Bk}, \frac{Bn}{k}]$, we have

$$\Pr_{T_i}[\mathcal{E}_{i,j} \wedge \neg \mathcal{E}_{i,j'}] = \Pr[\mathcal{E}_{i,j}] \cdot \Pr[\neg \mathcal{E}_{i,j'} \mid \mathcal{E}_{i,j}]$$

$$= \left(1 - \left(1 - \frac{|C_j|}{n}\right)^{|T_i|}\right) \cdot \left(1 - \frac{|C_{j'}|}{n}\right)^{|T_i|-1}$$

$$\geq \left(1 - \left(1 - \frac{1}{Bk}\right)^{k/B}\right) \cdot \left(1 - \frac{B}{k}\right)^{k/B} \geq \left(1 - \exp\left(B^{-2}\right)\right) \cdot \frac{1}{e} \geq \frac{1}{2eB^2}$$

and so for a fixed $(j, j') \in \binom{[k]}{2}$, we have

$$\Pr_{T_1, \ldots, T_s}[\forall i \in [s]\colon \neg\left(\mathcal{E}_{i,j} \wedge \neg \mathcal{E}_{i,j'}\right)] \leq \left(1 - \frac{1}{2eB^2}\right)^{2eB^2 \ln(100k^2)} \leq \frac{1}{100k^2}$$

and the claim follows by a union bound over all $(j, j') \in \binom{[k]}{2}$. $\qquad\qquad \square$

By Claim B.2, with probability at least $99/100$, for every $j \neq j' \in [k]$ we have some $T_i$ such that $T_i \cap C_j \neq \emptyset$ and $T_i \cap C_{j'} = \emptyset$. In particular, for $x \in U$, let $C_{j^*}$ be the cluster containing $x$. For every $j \neq j^*$ we have some $T_i$ such that $T_i \cap C_j \neq \emptyset$ and $T_i \cap C_{j^*} = \emptyset$ which means that in line (9) of the algorithm, we have $J_x = [k] \setminus \{j^*\}$ and so we successfully identify the cluster containing $x$. Moreover, this occurs for all $x$. Finally, line (4) succeeds with probability $99/100$ and thus the entire algorithm succeeds with probability at least $49/50$ by a union bound.

## C    Lower Bounds

### C.1    An $\Omega(\frac{n^2}{s^2})$ Lower Bound for Non-adaptive $3$-Partition Recovery

**Theorem C.1.** *Non-adaptive $3$-clustering requires $\Omega(n^2)$ pair queries.*

*Proof.* For every $(x, y) \in \binom{U}{2}$ consider the following pair of partitions:

$$P^1_{x,y} = (\{x, y\}, \emptyset, U \setminus \{x, y\}) \text{ and } P^2_{x,y} = (\{x\}, \{y\}, U \setminus \{x, y\}).$$

Observe that the oracle returns the same value for $P_{x,y}^1$ and $P_{x,y}^2$ on every possible query except on the set $\{x, y\}$. Thus, if query set $Q \subseteq U \times U$ distinguishes these two clusterings, then $Q \ni \{x, y\}$. Therefore, the number of pairs $\{x, y\}$ such that $Q$ distinguishes $P_{x,y}^1$ and $P_{x,y}^2$ is at most $|Q|$. Now, let $A$ be any non-adaptive pair-query algorithm which successfully recovers an arbitrary 3-clustering with probability $\geq 2/3$. The algorithm $A$ queries a random set $Q \subseteq U \times U$ according to some distribution, $\mathcal{D}_A$. In particular, for every $\{x, y\} \in \binom{U}{2}$, $Q$ distinguishes $P_{x,y}^1$ and $P_{x,y}^2$ with probability $\geq 2/3$. Thus,

$$\frac{2}{3} \binom{n}{2} \leq \sum_{\{x,y\} \in \binom{U}{2}} \Pr_{Q \sim \mathcal{D}_A} [Q \text{ distinguishes } P_{x,y}^1 \text{ and } P_{x,y}^2]$$

$$= \mathbf{E}_{Q \sim \mathcal{D}_A} \left[ \left| \left\{ \{x, y\} \in \binom{U}{2} : Q \text{ distinguishes } P_{x,y}^1 \text{ and } P_{x,y}^2 \right\} \right| \right] \leq |Q|$$

using linearity of expectation, and this completes the proof. □

**Corollary C.2.** *Non-adaptive* 3*-clustering requires* $\Omega(n^2/s^2)$ *subset queries of size at most* $s$.

*Proof.* This follows from Theorem C.1 since one $s$-sized query can be simulated by $\binom{s}{2}$ pair-queries. □

Thus, in order to achieve a near-linear non-adaptive upper bound for 3-clustering, we require an algorithm which makes queries of size $\widetilde{\Omega}(\sqrt{n})$.

### C.2 An $\Omega(n \log n)$ Lower Bound for Sample-Based 2-Partition Recovery

**Theorem C.3.** *Sample-based* 2*-clustering requires* $\Omega(n \log n)$ *subset queries.*

*Proof.* Let $|U| = n$ be even and let $A, B \subseteq U$ be two disjoint sets of size $|A| = |B| = n/2$. Let $P = (A, B)$ and for any $x \in U$ let $P_x$ denote the partition obtained by switching the set that $x$ belongs to. We show that it requires $\Omega(n \log n)$ sample-based subset queries to distinguish $P$ from $P_x$ for all $x$. For $x \in U$ and $T \subseteq U$, let $\mathcal{E}_{x,T}$ denote the event that querying $T$ distinguishes $P$ from $P_x$. Note that $\mathcal{E}_x$ occurs iff $x \in T$ and $T \setminus x \subseteq A$ or $T \setminus x \subseteq B$. Thus, for a random set $T$ of size $s \geq 2$, we have

$$\Pr_{T \,:\, |T|=s} \left[ \mathcal{E}_{x,T} \right] = s \cdot \frac{1}{n} \cdot 2 \cdot \left( \frac{n/2}{n} \right)^{s-1} = \frac{s}{n} \cdot \left( \frac{1}{2} \right)^{s-2} \leq \frac{2}{n} \tag{3}$$

since the second-to-last quantity is clearly maximized when $s = 2$. Now, let $Q$ be a collection of sets, each of which consists of some of number of independent uniform samples. Note that the cardinality of these sets can differ from one another. Note that $Q$ distinguishes $P$ from every $P_x$ iff $\mathcal{E}_{x,T}$ occurs for every $x$ and some $T$. By eq. (3) and a standard coupon-collector argument, if $|Q| = o(n \log n)$, then with high probability there will be some $x$ for which $\neg \bigvee_{T \in Q} \mathcal{E}_{x,T}$ occurs. □

## D Useful Lemmas

### D.1 Vector Support Recovery from OR Queries

Given $x \in \{0, 1\}^n$, let $\mathsf{supp}(x) = \{i : x_i = 1\}$ denote the support of $x$. An OR-query on set $S \subseteq [n]$ returns

$$\mathsf{OR}_S(x) = \bigvee_{i \in S} x_i = \mathbf{1} \left( \mathsf{supp}(x) \cap S \neq \emptyset \right).$$

This section discusses the problem of recovering the support of a vector via OR queries. In particular, we are interested in *non-adaptive* algorithms for this problem. The results in this section are standard in the combinatorial group testing and coin-weighing literature. See e.g. [26, 28] and also [48], who applied these results to obtain query algorithms for graph connectivity.

**Lemma D.1.** *Let* $x \in \{0, 1\}^n$ *such that* $|\mathsf{supp}(x)| = 1$. *There is a deterministic, non-adaptive algorithm that makes* $\lceil \log n \rceil$ OR *queries and returns* $\mathsf{supp}(x)$. *The runtime is also* $O(\log n)$.

*Proof.* Since $|\mathsf{supp}(x)| = 1$, an OR query on set $S$ is equivalent to taking $\langle x, v \rangle$ where $v_i = 1$ iff $i \in S$. Let $M$ be the $\lceil \log n \rceil \times n$ matrix whose $i$'th column is simply $b^i \in \{0, 1\}^{\lceil \log n \rceil}$, the binary representation of $i$. The rows of $M$ correspond to OR queries. Then, $Mx = \sum_{i=1}^{n} x_i b^i = \sum_{i:\, x_i=1} b^i = b_j$ where $j$ is the unique coordinate where $x_j = 1$. $\square$

**Lemma D.2.** *Let $x \in \{0, 1\}^n$. There is a deterministic, non-adaptive algorithm* SER1bit *that makes* $2\lceil \log n \rceil$ OR *queries and certifies whether* $|\mathsf{supp}(x)| = 0, |\mathsf{supp}(x)| = 1$, *or* $|\mathsf{supp}(x)| > 1$. *If* $|\mathsf{supp}(x)| = 1$, *then it outputs* $\mathsf{supp}(x)$. *The runtime is also* $O(\log n)$.

*Proof.* Let $M$ be the $\lceil \log n \rceil \times n$ matrix described in the proof of Lemma D.1. Let $\mathbf{1} = 1^{\lceil \log n \rceil \times n}$ denote the all 1's matrix with the same dimensions. We query $M \cdot x$ and $(\mathbf{1} - M) \cdot x$ where here $(\cdot)$ denotes the "OR product". I.e. the $i$'th coordinate of $M \cdot x$ is $\mathbf{1}((Mx)_i > 0)$. Note that $\mathbf{1} - M$ is obtained by flipping every bit in $M$. Note that if $|\mathsf{supp}(x)| = 1$, then $M \cdot x$ is guaranteed to return the unique coordinate where $x$ has a one, as in the proof of Lemma D.1. Thus, it suffices to show that we can use these queries to determine whether $|\mathsf{supp}(x)|$ is 0, 1, or strictly greater than 1.

First, $|\mathsf{supp}(x)| = 0$ iff $(M \cdot x)_1 = 0$ and $((\mathbf{1} - M) \cdot x)_1 = 0$ since the sets of 1-coordinates in the first row of $M$ and $\mathbf{1} - M$ partition $[n]$.

Next, we claim that $|\mathsf{supp}(x)| > 1$ iff there exists some $i \in [\lceil \log n \rceil]$ such that $(M \cdot x)_i = 1$ and $((\mathbf{1} - M) \cdot x)_i = 1$. Note that for every row $i$, the 1-coordinates in the $i$'th row of $M$ and $\mathbf{1} - M$ partition $[n]$. Thus, clearly if $(M \cdot x)_i = 1$ and $((\mathbf{1} - M) \cdot x)_i = 1$, then there are at least 2 coordinates where $x$ has a one. Now we prove the converse. Suppose there exists $i \neq j \in [n]$ where $x_i = x_j = 1$. Let $b^i, b^j \in \{0, 1\}^{\lceil \log n \rceil}$ denote the binary representations of $i, j$ respectively. Since $i \neq j$, there exists some bit $k$ where $b^i_k \neq b^j_k$. Without loss of generality let $b^i_k = 1$ and $b^j_k = 0$. Then,

$$(M \cdot x)_k = \mathbf{1}\left( \left( \sum_{\ell=1}^{n} x_\ell b^\ell \right)_k > 0 \right) = \mathbf{1}\left( \sum_{\ell:\, x_\ell=1} b^\ell_k > 0 \right) = 1,$$

$$((\mathbf{1} - M) \cdot x)_k = \mathbf{1}\left( \left( \sum_{\ell=1}^{n} x_\ell (\vec{1} - b^\ell) \right)_k > 0 \right) = \mathbf{1}\left( \sum_{\ell:\, x_\ell=1} (1 - b^\ell_k) > 0 \right) = 1$$

and this completes the proof. $\square$

Next, we describe a *randomized* non-adaptive algorithm for recovering the entire support of $x$.

**Lemma D.3.** *Let $x \in \{0, 1\}^n$. There is a non-adaptive algorithm that makes $O(t \log \frac{n}{\delta})$ OR queries on subsets of size $\lceil \frac{n}{t} \rceil$, and if $|\mathsf{supp}(x)| \leq t$, returns $\mathsf{supp}(x)$ with probability $1 - \delta$, and otherwise certifies that $|\mathsf{supp}(x)| > t$. The algorithm's runtime is $O(n \log \frac{n}{\delta})$.*

*Proof.* For brevity, we assume that $t$ divides $n$. Let $m = e \cdot t \ln \frac{n}{\delta}$. We make OR queries on sets $S_1, \ldots, S_m$, each formed by taking $n/t$ i.i.d. uniform samples from $[n]$ and define

$$X = [n] \setminus \bigcup_{\ell \in [m]:\, \mathsf{OR}_{S_\ell}(x)=0} S_\ell. \tag{4}$$

If $|X| > t$, we certify $|\mathsf{supp}(x)| > t$ and if $|X| \leq t$, then we output $X$.

Assuming a uniform sample from $[n]$ can be obtained in $O(1)$ time, the runtime of the algorithm is $O(m \cdot \frac{n}{t}) = O(n \ln \frac{n}{\delta})$.

Suppose that $|\mathsf{supp}(x)| > t$. Observe that $\mathsf{supp}(x) \subseteq X$ and so $|X| > t$ with probability 1. Thus, the algorithm is always correct in this case.

Now suppose $|\mathsf{supp}(x)| \leq t$. We argue that $X = \mathsf{supp}(x)$ with probability at least $1 - \delta$. Consider some $i \notin \mathsf{supp}(x)$. Note that $i \notin X$ iff there is some query $S_\ell \ni i$ for which $S_\ell \cap \mathsf{supp}(x) = \emptyset$. Let $\mathcal{E}_{i,\ell}$ denote the event that $i \in S_\ell$ and $S_\ell \cap \mathsf{supp}(x) = \emptyset$. Then, since $|\mathsf{supp}(x)| \leq t$, we have

$$\Pr[\mathcal{E}_{i,\ell}] = \frac{n}{t} \cdot \frac{1}{n} \cdot \left( 1 - \frac{|\mathsf{supp}(x)|}{n} \right)^{\frac{n}{t}-1} \geq \frac{1}{t}\left( 1 - \frac{t}{n} \right)^{\frac{n}{t}} \geq \frac{1}{et}$$

and so
$$\Pr[i \in X] = \Pr\left[\neg \mathcal{E}_{i,\ell} \text{ for all } \ell \in [m]\right] \leq \left(1 - \frac{1}{et}\right)^m \leq \frac{\delta}{N}$$

since $m = e \cdot t \ln \frac{N}{\delta}$. Thus, by a union bound, we have $\Pr[X \neq \mathsf{supp}(x)] \leq \delta$. $\qquad\square$

Finally, we make the following simple observation regarding algorithms that are restricted to making OR queries on subsets of bounded size.

**Observation D.4.** *A single* OR *query on a set $S$ can be simulated by $\frac{|S|}{s}$ queries of size at most $s$.*

Combining this observation with Lemma D.3 gives the following lemma.

**Lemma D.5.** *Let $x \in \{0,1\}^n$ and $s, t \geq 1$ be positive integers where $s \leq \frac{n}{t}$. There is a non-adaptive algorithm that makes $O(\frac{n}{s} \log \frac{n}{\delta})$* OR *queries on subsets of size $s$, and if $|\mathsf{supp}(x)| \leq t$, returns $\mathsf{supp}(x)$ with probability $1 - \delta$, and otherwise certifies that $|\mathsf{supp}(x)| > t$. The algorithm runs in time $O(n \log \frac{n}{\delta})$.*

### D.2 Connectivity of Erdös-Rényi Random Graphs

Our proofs in Section 2.1, Appendix A, and Appendix G make use of the following bound on the probability of a random graph being connected. For intuition, note that for sufficiently large $n$,

$$1 - (\delta/3n)^{2/n} \approx 1 - \exp(-\frac{2\ln(3n/\delta)}{n}) \approx \frac{\ln(3n/\delta)}{n}.$$

Thus, Fact D.6 asserts that for sufficiently large $n$ a random graph containing $\gg n \ln n$ edges is connected with high probability, which may be a more familiar statement to the reader. However, we need such a bound to be true even for very small $n$ and so we give the following more broadly applicable version.

**Fact D.6.** *Let $G(n,p)$ denote an Erdös-Rényi random graph. If $p \geq 1 - (\delta/3n)^{2/n}$, then $G(n,p)$ is connected with probability at least $1 - \delta$.*

*Proof.* A graph $G = (V, E)$ is connected if and only if for every cut $S \subset V$, there exists an edge $(u, v) \in E \cap (S \times \overline{S})$. When $G$ is drawn from $G(n,p)$, this does not occur for a cut $S$ of size $|S| = t$ with probability exactly $(1-p)^{t(n-t)}$. There are exactly $\binom{n}{t}$ such cuts. Thus, taking a union bound over all cuts and using our lower bound on $p$, we have

$$\Pr_{G \sim G(n,p)}[G \text{ not connected}] \leq \sum_{t=1}^{n-1} \binom{n}{t} \left(\frac{\delta}{3n}\right)^{\frac{2}{n} \cdot t(n-t)}$$

$$\leq 2 \sum_{t=1}^{\lfloor n/2 \rfloor} \binom{n}{t} \left(\frac{\delta}{3n}\right)^{\frac{2}{n} \cdot t(n-t)}$$

$$\leq 2 \sum_{t=1}^{\lfloor n/2 \rfloor} \binom{n}{t} \left(\frac{\delta}{3n}\right)^{\frac{2}{n} \cdot \frac{tn}{2}} \leq 2 \sum_{t=1}^{\lfloor n/2 \rfloor} n^t \left(\frac{\delta}{3n}\right)^t = 2 \sum_{t=1}^{\lfloor n/2 \rfloor} (\delta/3)^t \leq \delta$$

and this completes the proof. $\qquad\square$

## E An $O(n \log^2 k)$ Algorithm for the Balanced Case

In Appendix B, we gave an algorithm for $k$-clustering making $O(n \log k + k \log^4 k)$ subset queries when the cluster sizes are balanced within any constant factor. This query complexity simplifies to $O(n \log k)$ as long as $k = O(\frac{n}{\log^3 n})$. In this section we give an alternative algorithm which is more efficient when $k \gg \frac{n}{\log^3 n}$.

**Theorem E.1.** *There is a non-adaptive algorithm for recovering a $B$-balanced $k$-clustering using $O(B^2 n \log^2 k)$ subset queries of size $O(k)$ which succeeds with probability $99/100$.*

*Proof.* Recall that for a vector $v \in \{0, 1\}^n$, an OR query on a set $S \subseteq [n]$ returns $\mathsf{OR}_S(v) = \bigvee_{i \in S} v_i$. We will use the following lemma for recovering $\mathsf{supp}(v) = \{i \colon v_i = 1\}$ via OR queries. We prove and discuss this lemma in Appendix D.1 (see Lemma D.2).

**Lemma E.2.** *There is a deterministic, non-adaptive algorithm that takes an arbitrary $v \in \{0, 1\}^n$, makes $2\lceil \log n \rceil$ OR queries, and certifies whether $|\mathsf{supp}(v)| = 0, |\mathsf{supp}(v)| = 1$, or $|\mathsf{supp}(v)| > 1$. If $|\mathsf{supp}(v)| = 1$, then it outputs $\mathsf{supp}(v)$. The runtime is $O(\log n)$.*

Given $x \in U = \{x_1, \ldots, x_n\}$, let $C(x)$ denote the cluster containing it. Let $v^{(x)} \in \{0, 1\}^n$ denote the Boolean vector with $v_i^{(x)} = \mathbf{1}(x_i \in C(x))$. As in Section 2.2, we have $\mathsf{OR}_S(x) = \mathbf{1}(q(S \cup \{x\}) = q(S))$. I.e. OR queries to $v^{(x)}$ are simluted by two subset queries to the clustering. This implies the following corollary.

**Corollary E.3.** *Given a $k$-clustering on $U$ of size $n$ and an element $x \in U$, let $C(x)$ denote the cluster containing $x$. There is a deterministic non-adaptive algorithm which takes as input $x$ and a set $R \subseteq U$, makes $O(\log |R|)$ subset queries, and if $|R \cap C(x)| = 1$, then the algorithm returns the unique $z \in R \cap C(x)$, and otherwise certifies that $|R \cap C(x)| \neq 1$. The runtime is $O(\log |R|)$.*

The pseudocode for the algorithm is given in Alg. 5. In words, Corollary E.3 says that if we have a set $R$ containing exactly one representative from $C(x)$, then with $O(\log |R|)$ subset queries we can identify that representative. Thus, suppose we have a collection of sets $R_1, \ldots, R_s$ such that for every cluster $j \in [k]$, there is some $R_i$ containing a unique representative from $C_j$. Consider the bipartite graph where on the left we have $U$ and on the right we have $R_1 \cup \cdots \cup R_s$. Then, for every $x \in U$ and every $R_i$ we can run the procedure from Corollary E.3, and if it returns a representative $y \in R_i \cap C(x)$, then we add the edge $(x, y)$ to this graph. By the property of $R_1, \ldots, R_s$, two vertices $x, y \in U$ belong to the same cluster iff they are connected by a path of length 2 in this graph. We show that setting $s = \Theta(B^2 \log k)$ and letting each $R_i$ be a random sample of $k/B$ elements from $U$ results in a collection of sets with this good property with high probability. This leads to a query complexity of $n \cdot s \cdot O(\log k) = O(n \log^2 k)$.

---

**Algorithm 5:** Second Algorithm for the $B$-Balanced Case

1 **Input:** Subset query access to a $B$-balanced partition $C_1 \sqcup \cdots \sqcup C_k = U$ of $|U| = n$ points;
2 Choose $s = eB^2 \ln(100k)$ sets $R_1, \ldots, R_s$ each formed by $\frac{k}{B}$ uniform samples from $U$;
3 Construct a bipartite graph $G(U, \bigcup_{j=1}^s R_j, E)$ as follows;
4 **for** $x \in U$ and $i \in [s]$ **do**
5      Run the algorithm from Corollary E.3 on input $x$ and $R_i$;
6      **if** *the algorithm certifies there is a unique $y \in R_i$ such that $x, y$ are in the same cluster* **then**
7          Add the edge $(x, y)$ to $E(G)$;
8      **end**
9 **end**
10 Let $C_1, \ldots, C_\ell$ denote the connected components of $G$;
11 **Output** the clustering $(C_1, \ldots, C_\ell)$;

---

**Query complexity and time complexity:** The algorithm makes $n \cdot s \cdot O(\log \frac{k}{B}) = O(B^2 n \log^2 k)$ queries. We assume that a uniform random sample can be obtained in $O(1)$ time. Thus, line (2) runs in $O(Bk \ln k)$ time. By Corollary E.3, line (5) runs in time $O(|R_i|) = O(\log k)$. Thus, the entire for-loop (lines 4-9) runs in time $O(ns \log k) = O(B^2 n \log^2 k)$. The bipartite graph $G$ has at most $O(n + Bk \log k)$ vertices and at most $O(ns) = O(B^2 n \log k)$ edges. Thus, line (10) can be executed in time $O(B^2 n \log k)$ time. The total runtime is thus dominated by $O(B^2 n \log^2 k)$.

The correctness of the algorithm now follows immediately from the following claim. $\qquad \square$

**Claim E.4.** *With probability at least $99/100$, for every $j \in [k]$, there exists $i \in [s]$ such that $|R_i \cap C_j| = 1$.*

*Proof.* Fix $j \in [k]$ and $i \in [s]$. We have

$$\Pr[|R_i \cap C_j| = 1] = |R_i| \cdot \frac{|C_j|}{n} \cdot \left(1 - \frac{|C_j|}{n}\right)^{|R_i|-1} \geq \frac{k}{B} \cdot \frac{1}{Bk} \cdot \left(1 - \frac{B}{k}\right)^{k/B} \geq \frac{1}{eB^2}$$

and so for a fixed $j \in [k]$,

$$\Pr[\forall i \in [s] \colon |R_i \cap C_j| \neq 1] \leq \left(1 - \frac{1}{eB^2}\right)^{eB^2 \ln(100k)} \leq \frac{1}{100k}$$

and so by a union bound

$$\Pr[\exists j \in [k], \forall i \in [s] \colon |R_i \cap C_j| \neq 1] \leq \frac{1}{100}$$

and this completes the proof. $\qquad\square$

# F   Two-Round Algorithms

In this section we describe two algorithms that use *two rounds* of adaptivity. That is, these algorithms are allowed to specify a round of queries, receive the responses, perform some computation, then specify a second round of queries and receive the responses, before finally recovering the clustering. We give a simple *deterministic* algorithm making $O(n \log k)$ queries in Appendix F.1 and a randomized algorithm for recovering a balanced clustering with $O(n \log \log k)$ queries in Appendix F.2. Both algorithms exploit the additional round of queries to first compute a set containing exactly one representative from every cluster.

## F.1   A Two Round $O(n \log k)$ Deterministic Algorithm using Single Element Recovery

**Theorem F.1.** *There is a two-round, non-adaptive, deterministic algorithm for $k$-clustering using $O(n \log k)$ subset queries.*

---

**Algorithm 6:** Deterministic 2-Round Algorithm

---

**1 Input:** Subset query access to a hidden partition $C_1, \ldots, C_k$ of $U = \{x_1, \ldots, x_n\}$;
**2** *Round 1:*
**3 Query** $P_t = \{x_i : i \leq t\}$ for every $t \in [n]$;
**4** Define $R = \{x_t : q(P_t) - q(P_{t-1}) = 1\}$ containing exactly one point from every cluster;
**5** For each $y \in R$, define cluster $R_y = \{y\}$;
**6** *Round 2:*
**7 for** $x \in U$ **do**
**8**  $\quad$ Use the $O(\log k)$ deterministic non-adaptive algorithm of Corollary F.2 to find the unique
  $\quad\quad$ $y \in R$ for which $x, y$ lie in the same cluster;
**9**  $\quad$ Place $x$ into $R_y$;
**10 end**
**11 Output** clustering $(R_y : y \in R)$;

---

*Proof.* Pseudocode for the algorithm is given in Alg. 6. The runtime is clearly dominated by the for-loop (lines 7-9) which run in time $O(n \log k)$ by Corollary E.3. Fix an arbitrary ordering $U = \{x_1, \ldots, x_n\}$. The first round of queries (lines 3-5) is used to compute a set $R \subseteq U$ containing exactly one representative from every cluster. This is done by querying every prefix $P_t = \{x_1, \ldots, x_t\}$ and observing that $q(P_t) - q(P_{t-1}) = 1$ iff $x_t$ is the only representative for its cluster in $P_t$. Thus, the set $R$ computed in line (4) contains, for each cluster $C$, the first member of $C$ in the ordering $x_1, \ldots, x_n$. In particular, it contains exactly one representative from every cluster. The second round of queries is used to determine, for every $x \in U$, the unique representative of $C(x)$ in $R$ (see line 8). To accomplish this we recall Corollary E.3 from Appendix E which we restate below. This completes the proof. $\qquad\square$

**Corollary F.2.** *Given a $k$-clustering on $U$ of size $n$ and an element $x \in U$, let $C(x)$ denote the cluster containing $x$. There is a deterministic non-adaptive algorithm which takes as input $x$ and a set $R \subseteq U$, makes $O(\log |R|)$ subset queries, and if $|R \cap C(x)| = 1$, then the algorithm returns the unique $z \in R \cap C(x)$, and otherwise certifies that $|R \cap C(x)| \neq 1$.*

## F.2 A Two Round $O(n \log \log k)$ Algorithm for Balanced Clusters

Recall that a clustering $C_1 \sqcup \cdots \sqcup C_k = U$ is $B$-balanced if $\frac{n}{Bk} \leq |C_j| \leq \frac{Bn}{k}$.

**Theorem F.3.** *There is a two round, non-adaptive algorithm which recovers a $B$-balanced $k$-clustering using $O(\sqrt{B} \cdot n \log \log k)$ subset queries.*

*Proof.* We will use the following result of [49] on query-based reconstruction of bipartite graphs as a black-box. Given a bipartite graph $G(V, W, E)$, an edge-count query on $(S, T)$ where $S \subseteq V$, $T \subseteq W$ returns $|E \cap S \times T|$, the number of edges between $S$ and $T$.

**Lemma F.4** ([49], see Section 4.3). *There is a non-adaptive algorithm which reconstructs any bipartite graph $G(V, W, E)$ where (a) $|V| = n$, (b) $|W| = m$, and (c) every vertex in $V$ has degree at most $1$, using $O(n \cdot \frac{\log n}{\log m})$ edge-count queries.*

We will say a set $A \subseteq U$ is an *independent set* if each element of $A$ belongs to a distinct cluster. Given two independent sets $A, B$ let $M(A, B)$ be the matching where there is an edge from $x \in A$ to $y \in B$ if $x, y$ belong to the same cluster. We observe that edge-count queries in $M(A, B)$ can be simulated by subset queries, leading to the following corollary.

**Corollary F.5.** *Suppose that $A, B \subseteq U$ are independent sets. There is a deterministic, non-adaptive algorithm which reconstructs $M(A, B)$ using $O(|A| \cdot \frac{\log |A|}{\log |B|})$ subset queries.*

*Proof.* We need to show that an edge-count query $(S, T)$ where $S \subseteq A$, $B \subseteq T$ can be simulated by a constant number of subset queries. Let $m(S, T)$ denote the number of edges in $M(A, B)$ between $S$ and $T$. Since $A, B$ are independent sets, $S, T$ are also independent sets, and so we have

$$m(S, T) = q(S) + q(T) - q(S \cup T)$$

since $m(S, T)$ is the number of clusters intersected by both $S$ and $T$. Thus, one edge-count query to $M(A, B)$ can be simulated by three subset queries and this completes the proof. $\square$

Pseudocode for the algorithm is given in Alg. 7. The algorithm is parameterized in terms of a value $\tau > 1$ which we will choose later in the proof so as to minimize the query complexity. The first round is used to accomplish the following. In lines (4-5) we construct a set $R$ containing exactly one representative from every cluster and use this to define an initial clustering. In line (6) we sample random sets $I_1, \ldots, I_s$ and in line (8) make a query to each to check whether or not it is an independent set. Line (10) defines $V$ which is the union of all the $I_i$'s which are independent sets. We now describe the second round. In line (14) we run the procedure of Corollary F.5 to construct the matching $M(I_i, R)$ whenever $I_i$ is an independent set. Finally, we determine for every $x \in U$, the unique $y \in R$ for which $x, y$ belong to the same cluster. If $x \in V$ this is done in lines (18-20) by taking $x$'s neighbor in $M(I_i, R)$ for some independent set $I_i$. If $x \notin V$, this is done in lines (23-24) by running the procedure of Corollary F.2.

The algorithm always either outputs fail in line (11), or correctly reconstructs the clustering by Corollary F.5 and Corollary F.2. Thus we only need to argue that $|U \setminus V| \leq \frac{n}{\tau}$ occurs with probability at least $99/100$ allowing it to pass the check in line (11), and that conditioned on this, the algorithm makes $O(n \ln \ln k)$ queries when we set $\tau$ appropriately. Let us first count the number of queries conditioned on this event. Line (8) performs $s$ queries. Since each $I_i$ is of size $\sqrt{k}$ and $|R| = k$, by Corollary F.5, lines (13-14) perform a total of $O(s \cdot \sqrt{k} \ln \tau) = O(\sqrt{B} \cdot n \ln \tau)$ queries. Lines (22-23) use $|U \setminus V| O(\log k) = O(\frac{n}{\tau} \log k)$ queries. Setting $\tau = \Theta(\ln k)$ yields a query complexity of $O(\sqrt{B} n \log \log k)$. We now prove in Claim F.6 that the required bound on $|U \setminus V|$ holds with high probability, and this completes the proof. $\square$

**Claim F.6.** *With probability at least $99/100$, we have $|U \setminus V| \leq \frac{n}{\tau}$.*

*Proof.* We prove an appropriate bound on $\mathbf{E}[|U \setminus V|]$ and then apply Markov's inequality. Fix $x \in U$. For $i \in [s]$, let $\mathcal{E}_{x,i}$ denote the event that $x \in I_i$ and $I_i$ is an independent set. Observe that $x \in U \setminus V$ iff $\mathcal{E}_{x,i}$ does not occur for every $i \in [s]$. We first lower bound the probability of $\mathcal{E}_{x,i}$. Observe that

$$\Pr_{I_i}[\mathcal{E}_{x,i}] = \Pr[x \in I_i] \Pr[I_i \text{ an independent set } \mid x \in I_i] \tag{5}$$

**Algorithm 7:** Two Round Algorithm for Balanced Clustering

---

1  **Input:** Subset query access to a hidden partition $C_1 \sqcup \cdots \sqcup C_k = U$ of $|U| = n$ points;
2  *Round 1:*
3  **Query** $P_t = \{x_i \colon i \leq t\}$ for every $t \in [n]$;
4  Define $R = \{x_t \colon q(P_t) - q(P_{t-1}) = 1\}$ containing exactly one point from every cluster;
5  For each $y \in R$, define cluster $R_y = \{y\}$;
6  Sample $s = 10\sqrt{\frac{B}{k}} \cdot n \ln(100\tau)$ sets $I_1, \ldots, I_s \subset U$ each formed by $\sqrt{\frac{k}{10B}}$ samples from $U$;
7  **for** $i \in [s]$ **do**
8       **Query** $I_i$. (This is to check if $q(I_i) = |I_i|$, i.e. whether $I_i$ is an independent set.);
9  **end**
10 Let $V = \bigcup_{i \in [s] \colon q(I_i) = |I_i|} I_i$ be the points in $U$ lying in an independent set among $I_1, \ldots, I_s$;
11 If $|V| < n(1 - \frac{1}{\tau})$, then **output fail**. Otherwise, continue;
12 *Round 2:*
13 **for** $i \in s \colon q(I_i) = |I_i|$ **do**
14      Run the algorithm from Corollary F.5 on sets $I_i, R$ and let $M_i \subset I_i \times R$ be the output;
15 **end**
16 **for** $x \in U$ **do**
17      **if** $x \in V$ **then**
18          Choose $I_i$ such that $x \in I_i$ and $I_i$ is an independent set;
19          Let $y \in R$ denote the neighbor of $x$ in the matching $M_i \subset I_i \times R$;
20          Place $x$ into $R_y$;
21      **end**
22      **if** $x \in U \setminus V$ **then**
23          Use the $O(\log k)$ deterministic non-adaptive algorithm of Corollary F.2 to find the unique $y \in R$ for which $x, y$ lie in the same cluster;
24          Place $x$ into $R_y$;
25      **end**
26 **end**
27 **Output** clustering $(R_y \colon y \in R)$;

---

and

$$\Pr_{I_i}[x \in I_i] = 1 - \left(1 - \frac{1}{n}\right)^{|I_i|} \geq 1 - \exp\left(-\frac{|I_i|}{n}\right) \geq \frac{|I_i|}{2n} \geq \sqrt{\frac{k}{B}} \cdot \frac{1}{8n}$$

where we have used the inequality $\exp(-z) \leq 1 - \frac{z}{2}$ for $z \in [0,1]$. Next, by a simple union bound over all pairs in $I_i$ and the fact that every cluster is bounded as $|C_j| \leq \frac{Bn}{k}$, we have

$$\Pr[I_i \text{ not an independent set} \mid x \in I_i] \leq |I_i|^2 \frac{B}{k} \leq \frac{1}{10}.$$

Plugging these bounds back into Equation (5) yields $\Pr_{I_i}[\mathcal{E}_{x,i}] \geq \sqrt{\frac{k}{B}} \cdot \frac{1}{10n}$ and noting that these events are independent due to the $I_i$'s being independent yields

$$\Pr[x \notin V] = \Pr[\neg\mathcal{E}_{x,i}, \ \forall i \in [s]] \leq \left(1 - \sqrt{\frac{k}{B}} \cdot \frac{1}{10n}\right)^s = \exp(-\ln(100\tau)) = \frac{1}{100\tau} \quad (6)$$

where we have used the definition of $s = 10\sqrt{B/k} \cdot n \ln(100\tau)$. Finally, this implies $\mathbf{E}[|U \setminus V|] \leq \frac{n}{100\tau}$ and so by Markov's inequality $\Pr[|U \setminus V| > \frac{n}{\tau}] < \frac{1}{100}$. This completes the proof. $\qquad\square$

## G   Sample-Based Algorithm using Unbounded Queries

**Theorem G.1.** *There is a non-adaptive, sample-based $k$-clustering algorithm making $O(nk \log n)$ subset queries which is correct with probability at least $99/100$.*

*Proof.* The algorithm is defined in Alg. 8. The proof techniques are quite similar to that of Theorems 2.2 and A.1 detailed in Section 2.1 and appendix A. We also refer the reader to Section 2 for a discussion on the main ideas.

---

**Algorithm 8:** Sample-Based Algorithm Using Unbounded Queries

---

1 **Input:** Subset query access to a hidden partition $C_1 \sqcup \cdots \sqcup C_k = U$ of $|U| = n$ points;
2 *(Query Selection Phase)*
3 **for** $p = 0, 1, \ldots, \log n$ **do**
4      Initialize query set $Q_p \leftarrow \emptyset$;
5      **Repeat** $\frac{40nk \ln(300nk^2)}{2^p}$ times;
6      $\longrightarrow$ Sample $T \subseteq U$ formed by $2^p$ independent uniform sample from $U$;
7      $\longrightarrow$ **Query** $T$ and add it $Q_p$;
8 **end**
9 *(Reconstruction Phase)*
10 Initialize hypothesis clustering $\mathcal{C}_0 \leftarrow \emptyset$;
11 **for** $p = 0, 1, \ldots, \log n$ **do**
12      Let $\mathcal{C}_p$ denote the collection of clusters reconstructed before phase $p$;
13      Let $\mathcal{R}_p = \bigcup_{C \in \mathcal{C}_p} C$ denote the points belonging to these clusters;
14      Initialize $\mathcal{C}_{p+1} \leftarrow \mathcal{C}_p$;
15      Let $Q'_p = \{T \setminus \mathcal{R}_p : T \in Q_p \text{ and } |T \setminus \mathcal{R}_p| = 2\}$. Since each $T \in Q_p$ is a uniform random set, the elements of $Q'_p$ are uniform random pairs in $U \setminus \mathcal{R}_p$;
16      Let $Q''_p = \{\{x,y\} \in Q'_p : q(\{x,y\} = 1)\}$ denote the set of pairs in $Q'_p$ where both points lie in the same cluster. This set can be computed since $q(T \setminus \mathcal{R}_p) = q(T) - q(T \cap \mathcal{R}_p)$ and $q(T \cap \mathcal{R}_p)$ is known since at this point we have reconstructed the clustering on $\mathcal{R}_p$;
17      Let $G_p$ denote the graph with vertex set $U \setminus \mathcal{R}_p$ and edge set $Q''_p$;
18      Let $C_1, \ldots, C_\ell$ denote the connected components of $G_p$ with size at least $\frac{n}{2k \cdot 2^p}$;
19      Add $C_1, \ldots, C_\ell$ to $\mathcal{C}_{p+1}$;
20 **end**
21 **Output** clustering $\mathcal{C}_{\log n + 1}$

---

Since $\sum_{p=0}^{\log n} \frac{1}{2^p} = O(1)$, the number of queries made by the algorithm is $O(nk \log n)$. To prove correctness it suffices to prove the following lemma.

**Lemma G.2.** *For each $p = 0, 1, \ldots, \log n$, let $\mathcal{E}_p$ denote the event that all clusters of size at least $\frac{n}{2k \cdot 2^p}$ have been successfully recovered immediately following iteration $p$ of Alg. 8. Then,*

$$\Pr[\neg \mathcal{E}_0] \leq \frac{1}{100k} \quad \text{and} \quad \Pr[\neg \mathcal{E}_p \mid \mathcal{E}_{p-1}] \leq \frac{1}{100k} \quad \text{for all } p \in \{1, 2 \ldots, \log n\}.$$

The proof that Lemma G.2 implies Theorem G.1 is identical to the proof that Lemma 2.3 implies Theorem 2.2 given just after the statement of Lemma 2.3. Thus, we move on to proving Lemma G.2. $\qquad\square$

*Proof. of Lemma G.2.* First consider the case of $p = 0$. In this iteration, the algorithm queries $|Q_0| \geq 40 \cdot nk \ln(300nk^2)$ random pairs and we need to show that it successfully recovers all clusters with size at least $\frac{n}{2k}$ with probability at least $1 - \frac{1}{100k}$. Let $C$ denote any such cluster and recall from lines (16-17) the definition of the graph $G_0$ with vertex set $U$ and edge set $Q''_0$. We will show that the induced subgraph $G_0[C]$ is connected, and thus $C$ is correctly recovered in lines (18-19), with probability at least $1 - \frac{1}{100k^2}$. Since there are at most $k$ clusters, the lemma holds by a union bound.

Consider any two vertices $x, y \in C$ and note that $|Q_0| \geq \frac{20n^2 \ln(300nk^2)}{|C|}$ since $|C| \geq \frac{n}{2k}$. We lower bound the probability that $(x, y)$ is an edge in $G_0[C]$ as follows. Note that this occurs iff $\{x, y\} \in Q_0$. Using an identical calculation to that of eq. (1), this probability is at least $1 - (\frac{1}{300k^2|C|})^{2/|C|}$, implying that $G_0[C]$ is connected with probability at least $1 - \frac{1}{100k^2}$ by Fact 2.4.

The argument for the case of $p > 0$ is identical to the argument given in "Case 3" of in the proof of Lemma 2.3 in Section 2. $\qquad\square$

# H    An $O(n \log k)$ Adaptive algorithm

Here we sketch a simple adaptive algorithm using $O(n \log k)$ queries. Suppose, we have identified one element from $i$ clusters (initially $i = 0$, and we have $i \leq k$ always). Suppose they are $X = \{x_1, x_2, ..., x_i\}$. We now want to find the cluster to which a new point $y$ belongs to. We first query $\{X, y\}$. If the answer is $i + 1$, then $y$ is part of a new cluster and $i$ grows to $i + 1$. Otherwise, $y$ is part of the $i$ clusters, and we detect the cluster to which $y$ belongs to using a binary search. We consider the two sets $X_1 = \{x_1, x_2, .., x_{\lceil i/2 \rceil}\}$, and $X_2 = \{x_{\lceil i/2 \rceil + 1}, .., x_i\}$. We then query $\{X_1, y\}$. If the answer is $\lceil i/2 \rceil + 1$, then we search recursively in $X_2$, else if the query answer is $\lceil i/2 \rceil$, then we search recursively in $X_1$. Clearly, the query complexity is $O(\log k)$ per item, and it requires $O(\log k)$ rounds of adaptivity even to place one element.

