# OpenReview forum: "Clustering with Non-adaptive Subset Queries"
_NeurIPS.cc/2024/Conference — NeurIPS 2024 poster_

### Official Review · Reviewer_hP14 · 2024-06-30

**Soundness:** 4
**Presentation:** 3
**Contribution:** 4
**Rating:** 7
**Confidence:** 2

**Summary:**

**[Setting]**
This paper studies the problem of clustering $n$ items into $k$ clusters using an oracle that can tell how many ground-truth clusters are represented in any given subset $S$ of items. The goal is to develop non-adaptive algorithms (where all queries are chosen before the oracle answers anything) and study their sample complexity.

**[Contributions]**
1. Randomized non-adaptive algorithms for constant k that make:
    1. $O(n k \log \log n \log k)$ queries when subsets of any size can be queried
    2. $O(n k \log n \log\log n)$ queries when $|S| = O(\sqrt{n})$
2. Randomized non-adaptive algorithms for general k that make:
    1. $O(n \log n \log k (n/s + \log s))$ queries of size at most $s$
3. An $\Omega(\min(n^2 / s^2, n))$ lower bound for any non-adaptive algorithm that is allowed to query subsets of size at most $s$.

The developed algorithms run in polynomial time.

Additionally, the paper also presents results for the case when :
1. Clusters are roughly balanced (i.e., their sizes are within constant factors of each other). Here, the algorithm makes $O(n \log k)$ queries when $k = O(n/\log^3 n)$ and $O(n \log^2 k)$ queries for any $k$.
2. An adaptive round is allowed. This improves dependency on logarithmic factors.

For these results, the details are only in the appendices.

**Strengths:**

1. The paper is fairly comprehensive and presents query complexity results in a wide range of settings. The proposed algorithm are optimal up to logarithmic factors.
2. The algorithms are well motivated. I especially appreciate the high-level intuition because of the clarity it adds to the paper.
3. While I have some concerns about the problem setting (see Weaknesses), I believe the presented results are a very good first step towards filling an interesting research gap - non-adaptive algorithm for clustering with subset queries.

**Weaknesses:**

1. My primary concern is with the feedback model. The oracle returns the number of clusters in the subset $S$, which implies that it knows the correct grouping in $S$. Why then would it only return the number of clusters in $S$ in practice and not the clusters themselves? Perhaps there are bandwidth constraints on response in some applications? But wouldn't one require way fewer queries if the oracle returns the clusters in $S$, which alleviates the bandwidth concerns, especially in the non-adaptive setting?
2. This is a relatively minor concern, but the paper defers a lot of details to the appendix without even including high-level ideas. I recommend including some details in the main paper, perhaps at the cost of a few technical details from Section 2. For example,
    1. What changes to the algorithm allow the query complexity to improve for the roughly balanced case?
    2. What is the "extra round of adaptivity"?

**Questions:**

1. Please address point 1 under weaknesses.
2. It seems to me that both Algorithm 1 and 2 work for any value of k. Can the authors elaborate on what they mean by constant $k$ and general $k$? Do you have two algorithms because one of them has a better dependence on $k$ ($O(\log k)$ instead of $O(k \log k)$)? On a similar note, how should one go about choosing between Algorithm 1 and 2 in practice?
3. Your lower bound is based on the idea of converting subset queries to pairwise queries. Unfortunately, this hides the dependence on $k$. Are you aware of any lower bounds in the pairwise case that depend on $k$ as well?

**Limitations:**

While it is true that all theorems have their assumptions listed, it would help the readers if the authors include a dedicated paragraph pointing out some of the research gaps that are left open by their work.

---

> ### Author Rebuttal · Authors · 2024-08-06
>
> Thanks for this constructive evaluation. Please see the global response for the question related to motivation. The rest of the responses are provided below. We will also add some high level ideas from the appendix to the main paper as suggested.
>
> **Can the authors elaborate on what they mean by constant $k$ and general $k$**
>
> You are correct that both algorithms work for any value of $k$. We have two algorithms because one of them has better query complexity when $k$ is small and the other has better query complexity when $k$ is large. Specifically, the algorithm of Theorem 1.2 makes $O(n \log \log n \cdot k \log k)$ queries while Theorem 1.1 gives $O(n \log^2 n \log k)$. Thus, to be precise, Theorem 1.1 is better when $k \leq O(\frac{\log^2 n}{\log \log n})$ and Theorem 1.2 is better otherwise. In practice, the algorithm should be chosen based on where $k$ falls with respect to this threshold. We wrote "constant $k$" because we wanted to emphasize that Theorem 1.2 gives query complexity $O(n \log \log n)$ when $k$ is a constant and "general $k$" to emphasize that Theorem 1.1 gives query complexity $O(n \log^2 n \log k)$ for all $k$. We will add a few lines in the write up to make this more clear.
>
> **Are you aware of any lower bounds in the pairwise case that depend on $k$ as well?**
>
> For non-adaptive pairwise query algorithms there is an $\Omega(n^2)$ lower bound for $k=3$ which is as strong as possible (for any $k\geq 3$) since $O(n^2)$ is also a trivial upper bound for any $k$. For $k = 2$, $O(n)$ is possible non-adaptively and this is optimal since the algorithm will need at least one pairwise query involving every point. Thus, this is the complete picture for non-adaptive pairwise query algorithms.

---

> > ### Comment · Reviewer_hP14 · 2024-08-12
> >
> > Thank you taking time to respond to my questions. I appreciate the added motivation for subset queries and the clarification regarding constant-vs-general k. All the best for your submission !

---

### Official Review · Reviewer_WJzB · 2024-07-09

**Soundness:** 3
**Presentation:** 3
**Contribution:** 3
**Rating:** 7
**Confidence:** 3

**Summary:**

The problem of cluster recovery via membership queries asks to assign each point in a dataset to its $k$ ground-truth cluster by using as few queries as possible to an oracle. A well-studied oracle is the same-cluster oracle that answers, given two points, whether the points are in the same or in different clusters. The complexity of adaptive algorithms for this problem is $\Theta(nk)$, but non-adaptive algorithms require $\Omega(n^2)$ for $k \geq 3$. The authors of this paper consider subset queries to surpass this barrier. Given a set of points $S$, a subset query returns the number of clusters the points in $S$ belong to. It is known that the complexity of an adaptive algorithm is $O(n)$. The authors consider non-adaptive algorithm and obtain the following results.

For general subset queries, they obtain an algorithm that makes $O(n \log^2 n \log k)$ queries for unbalanced clusters, and $O(n \log k)$ queries for balanced clusters. For constant $k$, they obtain an algorithm that makes $O(n \log \log n)$ queries. If the allowed subset size of the queries is bounded by $s \leq \sqrt{n}$, their algorithms need an essentially optimal $\tilde{O}(n^2 / s^2)$ number of queries for constant $k$, and $\tilde{O}(n^2 / s)$ for arbitrary $k$. Finally, they show that 2-round adaptivity can improve on the log factors in the complexity.

The algorithms follow two algorithmic approaches. For constant $k$, the authors observe that one can use subset queries to identify an isolated point representing its cluster in a subset and explore the cluster from this point, or one can use subset queries that contain two points from the same cluster to sample intra-cluster edges. The two strategies have a reverse complexity with respect to the cluster size, and a combination of them give the final complexity trade-off. For general $k$, the authors related subset queries to Boolean OR queries, and use these to recover clusters one by one.

**Strengths:**

The paper provides a rather rich collection of results on foundational variants of the non-adaptive cluster recovery problem with near-linear complexity for unbounded subsets.

**Weaknesses:**

Unbounded subset size may be a rather strong assumption, and the results for bounded subset size of non-adaptive algorithms are not near-linear and almost trivial for constant subset size. One may be interested in few-round adaptive algorithms with bounded subset size.

**Questions:**

/

**Limitations:**

/

---

> ### Author Rebuttal · Authors · 2024-08-06
>
> Thank you for the very thoughtful review.
>
> **Unbounded subset size**
>
> We agree that unbounded subset size is a strong assumption and we would like to better understand the query complexity for bounded size in follow-up work. We have shown that $O(\frac{n^2}{s^2} k\log n)$ is possible non-adaptively with $s \ll \sqrt{n}$ and this is essentially optimal in terms of $n$ and $s$. It would be interesting to know if near-linear query complexity is possible with query sizes that are even smaller than $\sqrt{n}$, say ${\rm poly}\log(n)$. We know this is not possible non-adaptively, but your idea of studying bounded subset size algorithms with few rounds of adaptivity is a great idea to look at in follow-up work.

---

> > ### Comment · Reviewer_WJzB · 2024-08-08
> >
> > Thank you for your response! If I understand correctly, the details you provided agree with my understanding at large, and in particular, the bound is essentially optimal for "the largest possible" $s \in o(\sqrt{n})$ only (and not all $1 \leq s \ll \sqrt{n}$). If that's not the case, please follow up and feel free to let me know.

---

> ### Author Response · Authors · 2024-08-08
>
> Thank you for seeking clarification. We apologize if there were some ambiguities in the previous response.
>
> Our result is optimal for all values of $s \leq \sqrt{n}$ within a $\log n \log \log n$ factor.
>
> In particular, our algorithm in Theorem A.1 (or the simplified version, 1.4) achieves $O(\frac{n^2}{s^2} k \log n \log \log n)$ non-adaptive queries for any $s  \leq \sqrt{n}$. Thus for constant k (which is the regime for Theorem A.1 and 1.4), our bounds are optimal for all values of $s  \leq \sqrt{n}$ within the $O(\log n \log \log n)$ factor since $\frac{n^2}{s^2}$ is a lower bound for non-adaptive queries.
>
> So to summarize, it is not correct that the bound is optimal only for "the largest possible" $s \in o(\sqrt{n})$. Instead, the bound is optimal for all values of $s \leq \sqrt{n}$.

---

> > ### Comment · Reviewer_WJzB · 2024-08-14
> >
> > Sorry for the confusion, that was my bad, and thank you for the clarification!
> >
> > Edit note for review: Taking the whole rebuttal into account, I think this paper should be accepted (change score 6 -> 7).

---

### Official Review · Reviewer_SHzL · 2024-07-10

**Soundness:** 4
**Presentation:** 4
**Contribution:** 4
**Rating:** 8
**Confidence:** 4

**Summary:**

The paper studies the clustering problem with non-adaptive subset queries. The problem formulation is as follows. Suppose we are given an oracle $q: \mathcal{V}\rightarrow \mathbb{R}$ such that for any query on a subset $S\subseteq V$ of vertices, the oracle returns how many clusters are in $S$ in the optimal clustering. The goal is to recover the optimal clustering with subset queries that are as small as possible. The problem setting is similar to the clustering with same-cluster query problem, where the oracle returns whether two vertices $(u,v)$ are in the same cluster in the optimal clustering. Both problems are motivated by scenarios where we could use semi-supervised learning to obtain such types of oracles. However, in that setting, the number of necessary queries becomes $\Omega(n^2)$ when $k$ is at least $3$. As such, to circumvent this strong barrier, it is natural to look into the new model with extra power for the queries.

In the new model, we can easily obtain an algorithm $O(n\log{k})$ queries by binary search. However, such an algorithm requires at least $\Theta(\log{k})$ rounds of adaptivity. The paper focused on the non-adaptive setting, where the queries have to be made *independent* of the results of previous queries. The main results include $i).$ an algorithm that makes $O(n \log\log{n})$ queries when $k=O(1)$ (linear dependency on $k$), and; $ii).$ an algorithm that makes $O(n\log^{2}{n} \log{k})$ queries for general $k$. The paper also studied the case when the maximum size of $S$ is bounded and obtained tight bounds when ${|S|}\leq \sqrt{n}$.

**Techniques.** The two main algorithms used different ideas. For the algorithm with constant $k$, the key observation is that after recovering the biggest cluster(s) in the queried set, there are non-trivial probability that the remaining of the queried set has exactly one or two points. For the two cases, we can recover clusters via subset queries for some specific sizes. As such, we can repeat enough times, and use a ‘valid’ set to recursively recover the clusters in the reconstruction phase. For the algorithm with general $k$, the paper reduces the problem to solving the OR queries of boolean vectors. In this way, the paper provides some new results for OR queries and obtains the desired bound.

**Strengths:**

I have a positive view of the paper. In my opinion, the problem setting is well-motivated, and it is a natural extension of the well-studied same-cluster query model. The paper is well-written, and the authors did a nice job explaining the ideas in the high-level overview. Therefore, although the techniques are non-trivial, I managed to follow the ideas well. Given the page limits of the conference, the paper still managed to do fairly well in terms of organization.

**Weaknesses:**

I don’t see any major issue in the paper. Some minor issues to address:
- It might be good to further clarify what ‘non-adaptive’ means in your paper. I think in your algorithm, you could still adaptively ‘add on’ to clusters *after* making the queries. For a moment I thought your algorithm could conduct $\log {n}$ rounds *in parallel* to recover all clusters, which will be even stronger in the ‘non-adaptive’ notion.
- The paper doesn’t contain experiments. I’m personally OK with it, but I think for NeurIPS you might want to justify why this is not an issue. I think AKB [NeurIPS’16] (‘clustering with same-cluster queries’) gives a nice motivation for the same-cluster oracle model, and maybe this paper should provide some similar justifications.
- The dependence on $k$ should be mentioned after theorem 1.2. I think $O(nk)$ is really not that bad – I thought it was something like $2^k$.

**Questions:**

Most of the questions are asked in the ‘weakness’ part. Two MISC questions:
- Line 165: should it be $p\leq \log\log{n}$?
- If we just want the clustering of an $\alpha$ fraction of the vertices, can we use an even smaller number of queries?

**Limitations:**

I do not see any potential negative societal impacts.

---

> ### Author Rebuttal · Authors · 2024-08-06
>
> Thanks for this very encouraging message. Please see the responses to your evaluation below.
>
> **It might be good to further clarify what ‘non-adaptive’ means in your paper**
>
> Thank you, that is a good point. Our notion of non-adaptivity is only that the queries are made in one round. I.e. queries are not made based on responses from previous queries. Once the query responses are obtained there is no adaptivity-esque restriction on how the algorithm recovers the clusters. We agree that this should be clarified, and will do so in the future version.
>
> **Experiments**
>
> Being a theoretical paper, we thought simulations will not add a lot of value to the paper; however we can certainly add simulation of our algorithm.
>
> **The dependence on $k$**
>
> Thank you, that is a great point. We will add clarification regarding the dependence on $k$ directly after the theorem statement.
>
> **Line 165: should it be $p \leq \log \log n$?**
>
> No, here $p \leq \log n$ is correct since $\delta$ could be as small as $1/n$ and we need $2^p$ to approximate $1/\delta$ within a constant factor for some $p$. Note that in this paragraph we are only describing how this strategy leads to a $O(n \log n)$ query algorithm and the $O(n \log \log n)$ comes by combining the two strategies as described in the subsequent paragraphs.
>
> **If we just want the clustering of an $\alpha$ fraction of the vertices, can we use an even smaller number of queries?**
>
> This is a great question. Identifying the cluster containing any one point requires $\Omega(\log k)$ bits and so determining the clustering of $\alpha n$ points requires $\Omega(\alpha n \log k)$ bits. Since a query gives $O(\log k)$ bits of information we require $\Omega(\alpha n)$ queries to cluster an $\alpha$-fraction of the points. In particular, for any constant $\alpha$, the $\Omega(n)$ lower bound still holds.
>
> To answer your question directly, we do think there is a simple way to modify our algorithm to give improved query complexity in this case. For instance in "Strategy 1" described in line 161, we only need to handle the case of $\delta \geq \alpha/2$ since we can ignore clusters that are smaller than $\frac{\alpha}{2}n$ (we are talking specifically about the $k=3$ case, but this could be generalized). Thus we only need to iterate over $p \leq \log \frac{2}{\alpha}$ and this naturally leads to a $O(n \log \frac{1}{\alpha})$ query algorithm, which is $O(n)$ for constant $\alpha$. We will add this to the paper.
>
> It would be interesting to see if it is possible to get close to $O(\alpha n)$ for general $\alpha$, and we don't immediately see how to achieve this using our current ideas. Thank you for the great question.

---

> > ### Comment · Reviewer_SHzL · 2024-08-09
> >
> > Thanks for the clarifications and the response. I do not have any further questions.

---

### Official Review · Reviewer_MjFn · 2024-07-12

**Soundness:** 4
**Presentation:** 3
**Contribution:** 3
**Rating:** 6
**Confidence:** 4

**Summary:**

The paper gives results on clustering a dataset using "subset queries." This is a generalization of "same-cluster queries." The same cluster query asks whether two given elements belong to the same optimal cluster. The subset query asks how many different clusters the elements of a given subset span. There has been recent interest in clustering using same-cluster queries. This paper is an extension of this line of work. Clearly, O(n^2) queries are sufficient to cluster all the points. We would want to explore the possibility of making a much smaller number of queries.  There are two sub-models to consider when discussing subset queries:
1. Adaptive queries
2. Non-adaptive queries.
Adaptive queries allow the query algorithm to adapt new queries based on the response for the earlier queries, whereas, in the non-adaptive model, all subsets should be predefined before making any queries. Adaptive queries are more powerful since O(nk) adaptive same-cluster queries are sufficient, whereas O(n^2) are required in the non-adaptive setting. This paper gives results in the non-adaptive setting:
- An O(n log^2{n} log{k}) query algorithm for general values of k. This is using a reduction to a known problem in group testing. This is improved to O(n log{k}) for the balanced case (when cluster sizes are within a constant factor of each other).
- An O(n log log{n}) algorithm for constant values of k. This is using a simple randomized method.

**Strengths:**

- From a purely theoretical point of view, the results are interesting. The discussion is elaborate, with various special cases discussed. Except for logarithmic factors, the bounds are tight.
- The writing is good. The intuition is clearly stated within the main write-up, and the proofs are in the appendix.

**Weaknesses:**

- The motivation for considering subset queries is not immediately clear. The motivation for same-cluster queries was easier to understand since one can set up a crowdsource platform and ask users to check whether two objects (say images) belong to the same class. Setting up subset queries for large subsets may be tricky since answering such queries is not simple.

**Questions:**

- Subset queries may not be easy to answer accurately, specifically when the subsets are large. However, answering such queries approximately (within some margin of error) may be possible. It may be interesting to explore whether some interesting clustering can be obtained using a small number of such "inaccurate approximate queries."

**Limitations:**

This work is theoretical, and there are no potential negative societal impact.

---

> ### Author Rebuttal · Authors · 2024-08-06
>
> Thanks for the thoughtful evaluation of the paper. Please see the global response for the motivation of the model.
>
> **Subset queries may not be easy to answer accurately**
>
> This is a fantastic point. Clustering under noisy subset queries is a direction we're interested in exploring in follow-up work. Positive results under a meaningful/realistic noise model would create a more convincing justification for studying subset queries. We believe that our work is an important first step towards this.

---

> > ### Comment · Reviewer_MjFn · 2024-08-12
> > **Acknowledgement**
> >
> > Thanks for your response. I have no further questions. I will maintain my current score.

---

### Author Rebuttal · Authors · 2024-08-06

We sincerely thank all of the reviewers for their very thoughtful reviews and comments.

**Motivation for Subset Queries.**

As reviewer SHzL has pointed out, clustering with subset queries is a natural extension of the well-studied same-cluster queries. While with same-cluster queries we would require $O(n^2)$ non-adaptive queries to recover the clusters even for small $k$, here we show that just near-linear number of queries suffice with subset queries. To address the questions related to the practical motivation for the model, we provide a few justifications below. Rest of the questions have been addressed with a response to each reviewer separately.

- In the paper, we have considered non-adaptivity where queries are asked in one round as a major practical motivation. Several prior works have shown it is extremely time consuming to get query answers adaptively from a crowd. Subset queries is the only way to get good query complexity while considering non-adaptivity.
- As a reviewer pointed out, bounded size queries are more practical than unbounded size queries. We have studied the effect of query size for both of our algorithms. Our strategy 1 gives optimal dependency on query size.
- Our work serves as a stepping stone for future work to consider few rounds of adaptivity (partial answers have been provided in Appendix F), and where errors are allowed in query answers.
- As reviewer hP14 brings up, if there is a bandwidth constraint, then returning just a number is much easier. Suppose the algorithm sends its queries $S_1,\ldots,S_q$ off to $q$ different entities. Suppose these entities have a bandwidth constraint in that they do not want to send back much information. Answering a subset query requires them to send only $O(\log k)$ bits (just a number), instead of $O(|S_i| \log k)$ needed for describing the whole clustering on $S_i$. The total number of bits sent is $O(q \log k) = \widetilde{O}(n)$ and the max number of bits sent by any entity is $O(\log k)$.
 - If the model was instead that the query returned the entire clustering on $S$, then the problem is trivially solvable with $1$ query when unbounded size is allowed. On the other hand, for bounded size queries, it does not change the information theoretic lower bound.
 - Consider the problem of counting the number of clusters represented in $S$ vs computing the clustering in $S$ from the perspective of an entity answering a query. The counting problem is easier, because one does not have to resolve ambiguities in assignments. Therefore, even if assignments can be erroneous, counts are likely to be correct (hence more robust).


Again, thank you to all of the reviewers. We greatly appreciate the time and effort you've invested to carefully read and evaluate our work.

---

### Decision · Program_Chairs · 2024-09-25

**Decision:**

Accept (poster)

**Comment:**

This paper studies the problem of reconstructing a clustering by asking queries to the oracle, and proposes a new model for such interaction. The main weakness of the paper, mentioned by all four reviewers, is an unrealistically strong assumption that the oracle can return the exact count of clusters in an unbounded size query. This is a quite serious concern especially given that what we get from this strong assumptions (compared to the more standard pairwise same-cluster queries) is the ability to ask queries nonadaptiviely and to still get a nontrivial upper bound. I'm not entirely convinced by the arguments brought in the rebuttal, but the reviewers seem to be very positive about the paper regardless, so I lean towards acceptance.